# Cytosolic aggregation of mitochondrial proteins disrupts cellular homeostasis by stimulating the aggregation of other proteins

Urszula Nowicka[1,2,3], Piotr Chroscicki[2,4†], Karen Stroobants[5†], Maria Sladowska[2,4], Michal Turek[1,2,4], Barbara Uszczynska-Ratajczak[2,6], Rishika Kundra[5], Tomasz Goral[1,2], Michele Perni[5], Christopher M Dobson[5], Michele Vendruscolo[5], Agnieszka Chacinska[1,2,3]*

[1]ReMedy International Research Agenda Unit, University of Warsaw, Warsaw, Poland; [2]Centre of New Technologies, University of Warsaw, Warsaw, Poland; [3]IMol Polish Academy of Sciences, Warsaw, Poland; [4]International Institute of Molecular and Cell Biology, Warsaw, Poland; [5]Centre for Misfolding Diseases, Department of Chemistry, University of Cambridge, Cambridge, United Kingdom; [6]Institute of Bioorganic Chemistry, Polish Academy of Sciences, Poznan, Poland

**Abstract** Mitochondria are organelles with their own genomes, but they rely on the import of nuclear-encoded proteins that are translated by cytosolic ribosomes. Therefore, it is important to understand whether failures in the mitochondrial uptake of these nuclear-encoded proteins can cause proteotoxic stress and identify response mechanisms that may counteract it. Here, we report that upon impairments in mitochondrial protein import, high-risk precursor and immature forms of mitochondrial proteins form aberrant deposits in the cytosol. These deposits then cause further cytosolic accumulation and consequently aggregation of other mitochondrial proteins and disease-related proteins, including α-synuclein and amyloid β. This aggregation triggers a cytosolic protein homeostasis imbalance that is accompanied by specific molecular chaperone responses at both the transcriptomic and protein levels. Altogether, our results provide evidence that mitochondrial dysfunction, specifically protein import defects, contributes to impairments in protein homeostasis, thus revealing a possible molecular mechanism by which mitochondria are involved in neurodegenerative diseases.

**\*For correspondence:**
a.chacinska@imol.institute

†These authors contributed equally to this work

## Introduction

Although over 1000 proteins are utilized by mitochondria to perform their functions, only ~1 % of them are synthesized inside this organelle. The majority of mitochondrial proteins are synthesized in the cytosol and need to be actively transported to mitochondria, a process that occurs via a sophisticated system that involves protein translocases and sorting machineries (*Calvo et al., 2016*; *Morgenstern et al., 2017*; *Neupert and Herrmann, 2007*; *Pfanner et al., 2019*). The consequences of mitochondrial protein import defects on cellular proteostasis can be severe, and currently some response mechanisms are identified (*Boos et al., 2019*; *Izawa et al., 2017*; *Kim et al., 2016*; *Martensson et al., 2019*; *Priesnitz and Becker, 2018*; *Wang and Chen, 2015*; *Weidberg and Amon, 2018*; *Wrobel et al., 2015*; *Wu et al., 2019*; *Poveda-Huertes et al., 2020*). Mitochondrial dysfunction is closely associated with neurodegenerative disorders, and such mitochondrial defects as aberrant $Ca^{2+}$ handling, increases in reactive oxygen species, electron transport chain inhibition, and impairments

in endoplasmic reticulum-mitochondria tethering are well-described pathological markers (*Cabral-Costa and Kowaltowski, 2020*). Still unknown, however, is whether mitochondrial defects appear as a consequence of neurodegeneration, whether they contribute to it, or whether both processes occur. Disease-related proteins can interfere with mitochondrial import and the further processing of imported proteins within mitochondria (*Cenini et al., 2016*; *Di Maio et al., 2016*; *Mossmann et al., 2014*; *Vicario et al., 2018*). Furthermore, aggregated proteins can be imported into mitochondria where they can be either cleared or sequestered in specific deposit sites (*Bruderek et al., 2018*; *Ruan et al., 2017*; *Sorrentino et al., 2017*). However, the reverse aspect of the way in which mitochondrial dysfunction, including mitochondrial import defects, contributes to the progression of neurodegenerative diseases remains elusive. One possible mechanism may occur through alterations of cellular homeostasis as mitochondrial dysfunction can affect it through multiple mechanisms (*Andreasson et al., 2019*; *Braun and Westermann, 2017*; *Escobar-Henriques et al., 2020*).

Impairments in mitochondrial protein import and mitochondrial import machinery overload result in the accumulation of mitochondria-targeted proteins in the cytosol and stimulation of mitoprotein-induced stress (*Boos et al., 2019*; *Wang and Chen, 2015*; *Wrobel et al., 2015*). These findings raise the issue of whether the accumulation of mistargeted mitochondrial proteins contributes to the progression of neurodegenerative diseases. Additionally, unknown are whether mitoprotein-induced stress is a general response to precursor proteins that globally accumulate in the cytosol and whether a subset of mitochondrial precursor proteins pose particularly difficult challenges to the protein homeostasis system and consequently contribute to the onset and progression of neurodegenerative disorders (*Boos et al., 2020*; *Mohanraj et al., 2020*).

The analysis of a transcriptional signature of Alzheimer's disease supports the notion that there is a subset of mitochondrial proteins that is more dangerous than others for the cell (*Ciryam et al., 2016*; *Kundra et al., 2017*). These studies have shown that specific mitochondrial proteins that are functionally related to oxidative phosphorylation are transcriptionally downregulated in Alzheimer's disease. In the present study, we investigated why these proteins are downregulated. We hypothesized that this need arises from the potential supersaturation of these proteins, which makes them prone to aggregation (*Ciryam et al., 2016*; *Kundra et al., 2017*). Our results showed that when some of these mitochondrial proteins remain in the cytosol because of mitochondrial protein import insufficiency, they formed insoluble aggregates that disrupted protein homeostasis. These proteins triggered a prompt-specific molecular chaperone response that aimed to minimize the consequences of protein aggregation. However, when this rescue mechanism was insufficient, these aggregates stimulated the cytosolic aggregation of other mitochondrial proteins and led to the downstream aggregation of non-mitochondrial proteins. Our findings indicate that metastable mitochondrial proteins can be transcriptionally downregulated during neurodegeneration to minimize cellular protein homeostasis imbalance that is caused by their mistargeting.

## Results

### Metastable mitochondrial precursor proteins can aggregate in the cytosol

The analysis of a transcriptomic signature of Alzheimer's disease identified oxidative phosphorylation as a pathway that is metastable and downregulated in the human central nervous system (*Ciryam et al., 2016*; *Kundra et al., 2017*). This observation suggests that a group of mitochondrial proteins might be dangerous for cellular protein homeostasis because of their poor supersaturation and hence solubility at cellular concentrations. From the list of genes that were simultaneously downregulated and metastable in Alzheimer's disease patients, we selected all genes that encode mitochondrial proteins. Next, we identified genes that encode proteins that have homologs in yeast (*Figure 1—source data 1*). Based on the yeast homolog sequence, we generated FLAG-tagged constructs that were expressed under control of the copper-inducible promoter (CUP1). We then established a multi-centrifugation step assay to assess whether these proteins exceed their critical concentrations and become supersaturated when overproduced (*Vecchi et al., 2020*), thereby acquiring the ability to aggregate during their trafficking to mitochondria (*Figure 1—figure supplement 1A*). We followed a FLAG peptide signal to determine whether the protein was present in the soluble ($S_{125k}$) or insoluble ($P_{125k}$) fraction. We found that the β and g subunits of mitochondrial $F_1F_O$ adenosine triphosphate (ATP)

synthase (Atp2 and Atp20, respectively) were present in the insoluble fraction, indicating that they formed high-molecular-weight deposits (*Figure 1A*, *Figure 1—figure supplement 1B*). We made a similar observation for Rieske iron-sulfur ubiquinol-cytochrome *c* reductase (Rip1). Rip1 and subunit VIII of cytochrome *c* oxidase complex IV (Cox8) had entirely insoluble precursor (p) forms, whereas the mature (m) forms were partially insoluble (the p form of the protein is larger and migrates on the gel above the m form; *Figure 1A*, *Figure 1—figure supplement 1C*). Furthermore, the mature forms of subunit VIII of ubiquinol cytochrome *c* reductase complex III (Qcr8), core subunit of the ubiquinol-cytochrome *c* reductase complex (Cor1), and subunit β of the mitochondrial processing protease (MPP; Mas1) were partially insoluble. Only subunit VIb of cytochrome *c* oxidase (Cox12) and subunit 6 of ubiquinol cytochrome *c* reductase complex (Qcr6) were mainly present in the soluble fraction (*Figure 1A*, *Figure 1—figure supplement 1C*). Moreover, the tendency to aggregate and its harmful consequences correlated well with growth defects of yeast transformants that overexpressed mitochondrial proteins (*Figure 1B*, *Figure 1—figure supplement 1D*). We observed that the higher amount of an overproduced protein in the insoluble fraction correlated with a greater increase in lethality. This difference was still present under heat shock conditions of 37 °C, indicating that the general molecular chaperone response that was associated with an increase in temperature did not compensate for the observed changes for Atp2, Cox8, or pRip1 and could only partially compensate for Cox12 and Atp20.

To test whether these metastable mitochondrial proteins aggregate in the cytosol as a result of an inefficient mitochondrial protein import system, we followed the cytosolic fate of non-imported mitochondrial proteins that contained a cleavable targeting sequence. Consistent with previous studies (*Wrobel et al., 2015*), cells that were treated with the chemical uncoupler carbonyl cyanide m-chlorophenyl hydrazine (CCCP) had a compromised mitochondrial inner membrane (IM) electrochemical potential and protein import failure (*Chacinska et al., 2009*; *Neupert and Herrmann, 2007*; *van der Laan et al., 2010*). To observe import defects without prompting negative consequences of CCCP (e.g., autophagy), the treatment time was limited to 30 min. Perturbations of IM potential resulted in the accumulation of p forms of Rip1 (pRip1), similar to when the full precursor form of Rip1 was overproduced (*Figure 1C and D*, *Figure 1—figure supplement 1E*). We also extended the analysis of p forms of other proteins for which antibodies were available and allowed p form detection, namely mitochondrial matrix superoxide dismutase (Sod2) and mitochondrial malate dehydrogenase 1 (Mdh1) (*Figure 1C*, *Figure 1—figure supplement 1E*). The accumulation of pRip1, the p form of Sod2 (pSod2), and the p form of Mdh1 (pMdh1) increased as CCCP concentrations increased (*Figure 1C and D*). Next, an aggregation assay was performed to assess the fate of precursor mitochondrial proteins under conditions of chemical impairments in mitochondrial import. pRip, pSod2, and pMdh1 were present in the pellet fractions, demonstrating that even without overproduction they formed insoluble aggregates in the cell (*Figure 1E*). We also tested conditions of defective protein import by using a temperature-sensitive mutant of TIM23 presequence translocase, *pam16-3* (*Wrobel et al., 2015*). pRip1 and pSod2 accumulated (*Figure 1—figure supplement 1F*) and consequently formed insoluble aggregates under these conditions (*Figure 1F*). Therefore, the mistargeted precursor forms of mitochondrial proteins aggregated, when their concentrations exceeded their solubility limits due to their overproduction, when the mitochondrial import defect was stimulated chemically, and when mutants of the presequence translocase import pathway were used.

## Mitochondrial protein aggregation stimulates a cytosolic molecular chaperone response

To further investigate the cellular response to the accumulation of mitochondrial precursor forms in the cytosol, we investigated global transcriptomic changes that were triggered by pRip1 overproduction and *pam16-3* mutation (*Figure 2—source data 1*, *Figure 2—source data 2*, *Figure 2—source data 3*, *Figure 2—source data 4*, *Figure 2 A-C* , *Figure 2—figure supplements 1 and 2*). pRip1 overexpression triggered small and rather specific transcriptomic changes, in which a total of nine genes (*Figure 2C*) were upregulated and four were downregulated (see also *Figure 2A*, *Figure 2—figure supplement 1A*). The Kyoto Encyclopedia of Genes and Genomes (KEGG) enrichment analysis indicated that the oxidative phosphorylation pathway was upregulated, as expected, for pRip1 overexpression (*Figure 2—figure supplement 1B*). After 4 hr of pRpi1 expression, the ATP-binding cassette (ABC) transporters appeared as the second most upregulated KEGG pathway (*Figure 2—figure*

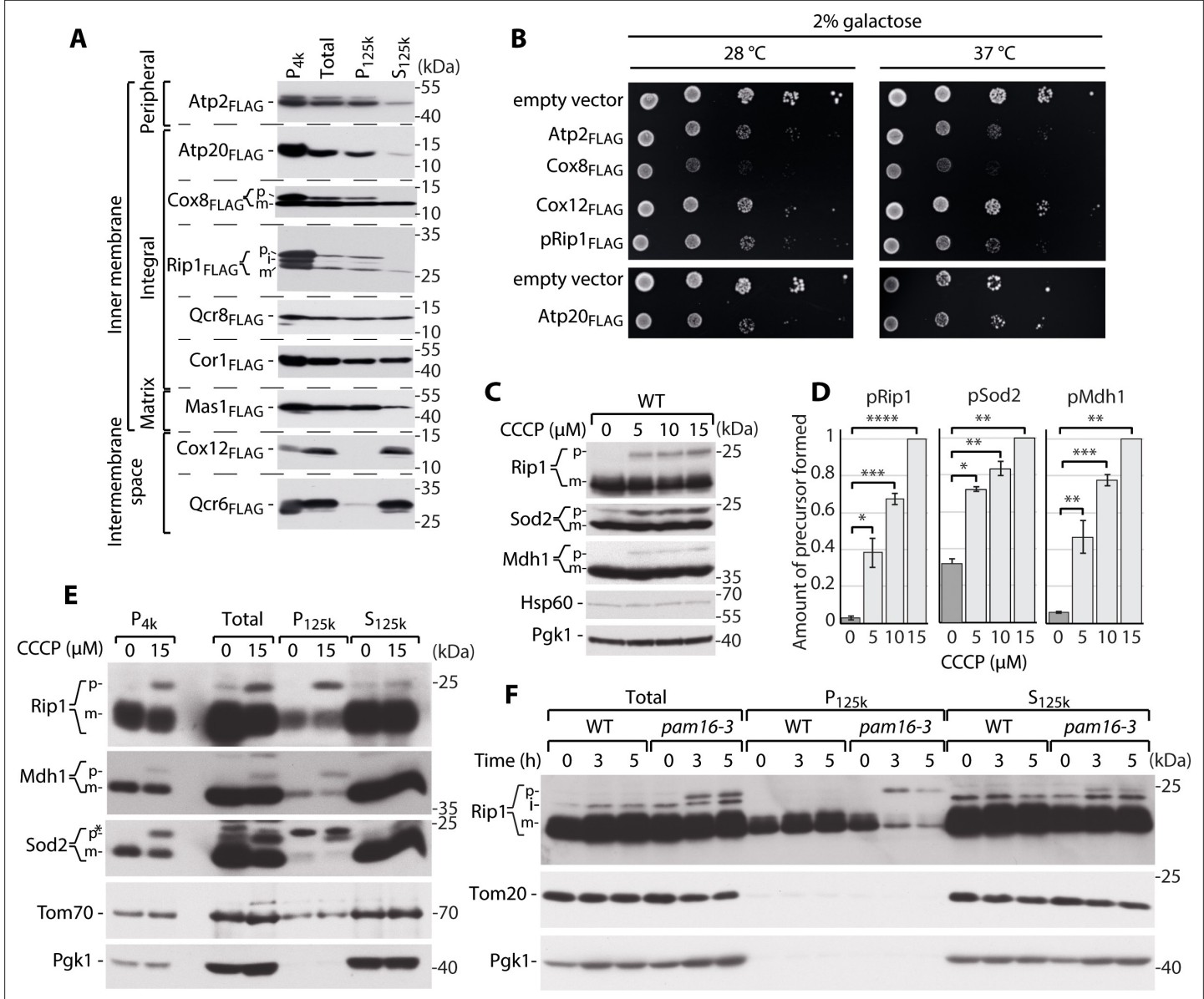

**Figure 1.** Supersaturated nuclear-encoded mitochondrial proteins aggregate in the cytosol. (**A**) SDS-PAGE analysis of aggregation assay fractions of WT yeast cells that overexpressed Atp2$_{FLAG}$, Atp20$_{FLAG}$, Cox8$_{FLAG}$, pRip1$_{FLAG}$, Qcr8$_{FLAG}$, Cor1$_{FLAG}$, Mas1$_{FLAG}$, Cox12$_{FLAG}$, and Qcr6$_{FLAG}$ for 3 hr when 2 % galactose with 0.1 % glucose was used as the carbon source. Pellet fractions after 4000 and 125,000× $g$ centrifugation are indicated as P$_{4k}$ and P$_{125k}$, respectively. The soluble fraction at 125,000× $g$ is indicated as S$_{125k}$. n = 3. (**B**) Ten-fold dilutions of WT cells that expressed metastable proteins or controls that were spotted on selective medium agar plates with galactose as the main carbon source at 28 and 37°C. (**C**) Total protein cell extract from WT yeast grown at 24 °C and treated with 0, 5, 10, or 15 µM carbonyl cyanide m-chlorophenyl hydrazine (CCCP) for 30 min. (**D**) Quantification of pRip1, pSod2, and pMdh1 from (**C**). Quantified data are shown as mean ± SEM. n = 3. (**E**) SDS-PAGE analysis of aggregation assay fractions of yeast cells that were treated with 15 µM CCCP for 30 min, with 2 % sucrose as the carbon source. (**F**) SDS-PAGE analysis of aggregation assay fractions of WT (*pam16WT*) and *pam16-3* mutant yeast strains grown at 19 °C and shifted to 37 °C for 0, 3, or 5 hr, with 2 % sucrose as the carbon source. In (**A**), (**C**), (**E**), and (**F**), the samples were separated by SDS-PAGE and identified by western blot with specific antisera. n = 3 for each experiment. *Nonspecific; p: presequence protein; i: intermediate protein; m: mature protein. *p<0.05, **p≤0.01, ***p≤0.001, ****p≤0.0001.

The online version of this article includes the following source data and figure supplement(s) for figure 1:

**Source data 1.** Aggregation propensity characterization of mitochondrial proteins.

**Figure supplement 1.** Supersaturated nuclear-encoded mitochondrial protein aggregate in the cytosol and stimulate growth defects.

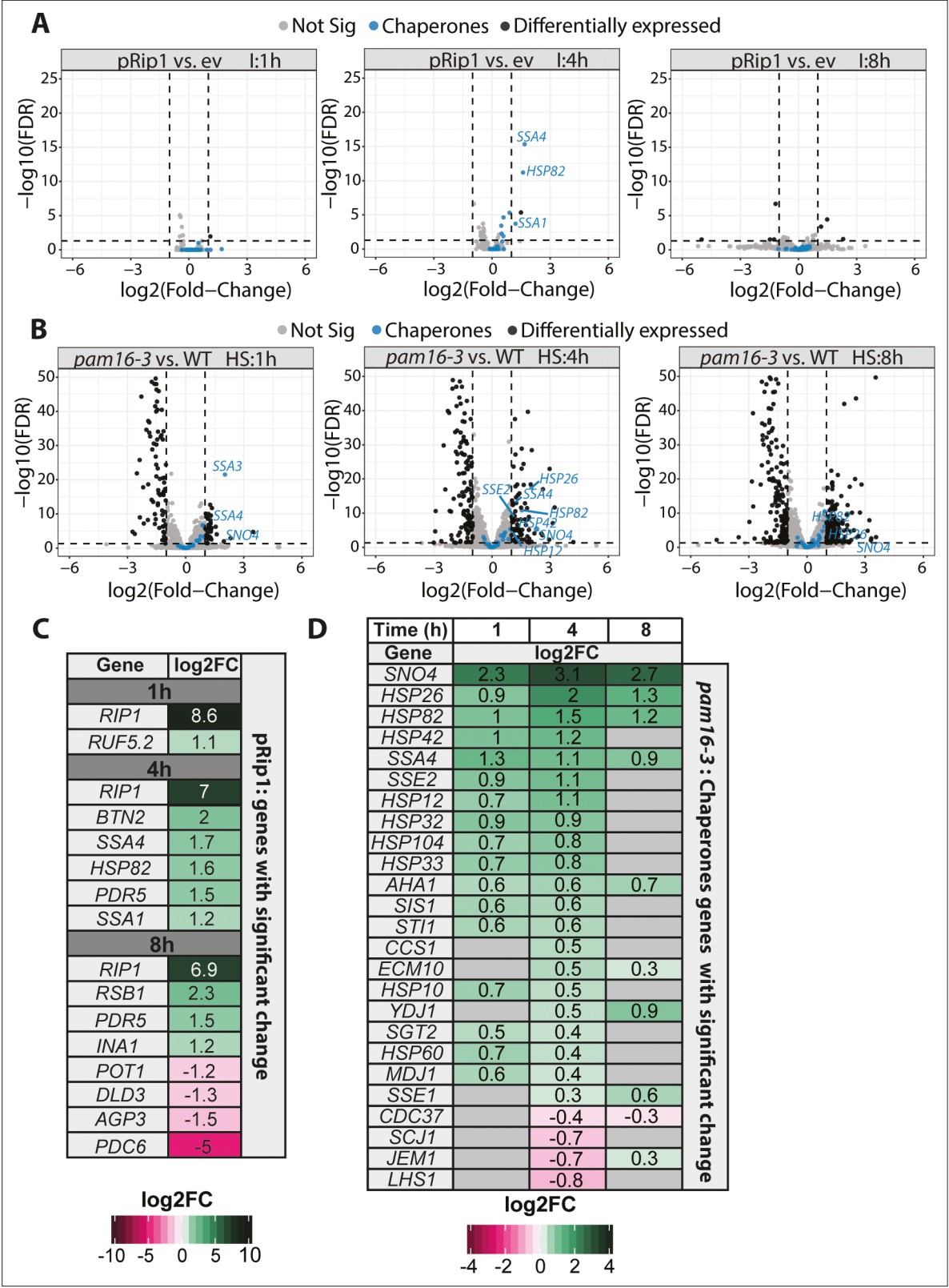

**Figure 2.** Mitochondrial protein import defects stimulate specific transcriptomic responses. (**A, B**) Volcano plot comparison of changes in expression (in terms of the logarithm of the fold change) assessed by the RNA-seq analysis of (**A**) WT yeast cells that overexpressed pRip1 compared with the empty vector (ev) control (induction [I] was performed under control of the CUP1 promoter for 1, 4, or 8 hr for both ev and pRip1) and (**B**) the *pam16-3* strain compared with WT (*pam16WT*) (both cell types were grown at a permissive temperature of 19 °C and shifted to a restrictive temperature of 37 °C for

*Figure 2 continued on next page*

*Figure 2 continued*

1, 4, or 8 hr). HS: heat shock time (in hours). Differentially expressed genes that encode molecular chaperones are indicated in blue. The gene name is displayed for each molecular chaperone if it was detected as differentially expressed (i.e., 5 % false discovery rate [FDR], log2 fold change [log2FC] ± 1). (**C**) All differentially expressed genes in pRip1 samples after 1, 4, and 8 hr of overexpression. (**D**) Analysis of changes in the expression of genes that encode molecular chaperones using *pam16-3* samples that were treated as in (**B**). Up- and downregulated genes (5 % FDR) are shown in green and pink, respectively. The intensity of the color shades reflects the level of expression change (log2FC). Genes that were not detected or those without statistically significant expression changes are depicted in gray.

The online version of this article includes the following figure supplement(s) for figure 2:

**Source data 1.** Full list of gene changes in response to pRip1 overexpression.

**Source data 2.** Full list of gene changes in response to *pam16-3* overexpression.

**Source data 3.** Gene expression matrices in response to pRip1 overexpression.

**Source data 4.** Gene expression matrices in response to *pam16-3* mutation.

**Figure supplement 1.** Characterization of gene change that was attributable to mitochondrial protein import defects.

**Figure supplement 2.** Characterization of gene change that was attributable to heat shock in WT and the pam16-3 mutant.

---

*supplement 1B*). For pRip1, the largest fold change was observed for the *SSA1*, *SSA4*, and *HSP82* chaperones as a likely response to pRip1 aggregation (***Figure 2A and C***). Extended pRip1 overproduction started to affect cellular performance as the following genes became downregulated: *POT1* (a peroxisomal protein that is also localized to the mitochondrial IM space and involved in fatty acid β-oxidation), *DLD3* (involved in lactate biosynthesis), *AGP3* (amino acid transmembrane transporter), and *PDC6* (pyruvate decarboxylase) (***Figure 2C***).

Next, since we planned to use a temperature-sensitive allele, we wanted to exclude possible effects that arise merely from a temperature shift. For that purpose, we analyzed the effect of heat shock individually on wild-type *pam16* (referred to as *WT*) and the *pam16-3* mutant. For WT, the increase in temperature resulted in a consistent drop in the amount of up- and downregulated genes (***Figure 2— figure supplement 2A***). For the *pam16-3* mutant, the number of up- and downregulated genes was nearly constant (***Figure 2—figure supplement 2B***), suggesting an unchanging need for adaptation. To determine the function of genes whose expression changed in response to the temperature shift, we performed KEGG enrichment analysis for WT and the *pam16-3* mutant. In both cases, we observed the upregulation of genes that are associated with ribosomes and ribosome biogenesis (***Figure 2— figure supplement 2C and D***). Unsurprisingly, the longer exposure of *pam16-3* yeast cells to 37 °C also affected metabolic pathways (***Wrobel et al., 2015***).

Finally, we studied effects of *pam16-3* mutations alone. For each heat shock time, *pam16-3* was compared with WT at the same time of exposure to 37 °C. With regard to mitochondrial defects, the *pam16-3* mutation had a stronger effect on the transcriptome than the effect of pRip1 (***Figure 2— figure supplement 1A***). Based on the KEGG analysis and as expected, oxidative phosphorylation-, metabolic pathway-, and citrate cycle-related genes were downregulated (***Figure 2—figure supplement 1C***). More than half of the genes that were affected encode mitochondrial proteins (***Morgenstern et al., 2017***; ***Figure 2—figure supplement 1D***). We also tested whether any groups of genes had related functions but were not enriched in the KEGG analysis. We analyzed all genes that passed a 5 % false discovery rate (FDR) cut-off and were non-mitochondrial. From this list, we selected genes for which we could identify at least 10 that shared similar functions. Based on this analysis, we noticed that a group of genes that encode molecular chaperones was upregulated in the *pam16-3* mutant (***Figure 2D***). We observed the upregulation of *HSP82* and *SSA4*, which were also upregulated in the case of pRip1 overexpression. We also observed the upregulation of other molecular chaperone genes, including *SNO4*, *HSP26*, *HSP42*, *SSE2*, *HSP12*, *HSP32*, *HSP104*, and *HSP33* (***Figure 2B***, blue, and ***Figure 2D***). The effect of the *pam16-3* mutant could be both direct and indirect when considering the various pathways in which these molecular chaperones are involved. We used the temperature-sensitive *pam16-3* mutant; therefore, we investigated the degree to which mitochondrial dysfunction exceeded the change that was attributable to heat shock treatment alone for each of these molecular chaperones. We compared changes in WT samples at both 19 and 37°C and matched them to corresponding changes that were attributable to the *pam16-3* mutation (normalized to WT at the same heat shock time) (***Figure 2—figure supplement 1E*** and ***Supplementary file 1***). Our analysis showed that the molecular chaperone response in *pam16-3* went beyond the changes that were expected

solely for the high-temperature treatment. The most pronounced changes were observed for *SSE2*, *SIS1*, *HSP42*, and *HSP104* at 4 hr of treatment (*Figure 2—figure supplement 1E*, black arrows). These findings suggest that impairments in mitochondrial protein import that originate from defects of translocases had to some extent similar, but not identical, consequences as clogging the import sites (*Boos et al., 2019*). To compare the pRip1 data with the effects of the *pam16-3* mutation, we investigated whether ABC transporter genes would also be upregulated. The analysis of significantly altered genes showed that as the heat shock time was extended, *PDR5* gene levels changed, as well as other ABC transporter genes, including *PDR15* and *SNQ2* (*Figure 2—figure supplement 1F*). *PDR5* is controlled by PDR3 transcription factor, the levels of which also increased (*Gulshan et al., 2008*). All of these genes were shown to be involved in the mitochondrial compromised protein import response, mitoCPR response (*Weidberg and Amon, 2018*). When we extended our analysis to other mitoCPR-related genes, we observed their upregulation beginning at 4 hr of treatment (*Figure 2—figure supplement 1F*).

## Effects of the Hsp42 and Hsp104 chaperone response to mitochondrial import failure at the protein level

Next, we assessed whether the observed molecular chaperone upregulation would also be observed at the protein level. We used CCCP to stimulate mitochondrial dysfunction to avoid the temperature factor at initial screening. We monitored changes in protein levels for all molecular chaperones against which we had antibodies available at the time of the studies, including Hsp104, Ssa1, Ssb1, Ssc1, Hsp60, and Edg1. No changes were observed for most of them, but a significant response was observed for Hsp104 (*Figure 3A and B*). Notably, Hsp104 upregulation at the protein level was observed already after 15 min of treatment (*Figure 3—figure supplement 1A*). We then tested whether Hsp104 was upregulated in response to mitochondrial import defects that were caused by the *pam16-3* mutation, along with the established *pam16-1* and *pam18-1* mutants and mitochondrial intermembrane space import and assembly (MIA) import pathway mutations *mia40-4int* and *mia40-3* (*Wrobel et al., 2015*). For all of the mutants under permissive conditions (19 °C; i.e., when only a mild protein import induction phenotype occurs), we observed the upregulation of Hsp104 at the protein level (*Figure 3C*). This effect was still present at the restrictive temperature of 37 °C, but the differences were less evident because of the activation of heat shock responses (*Figure 3—figure supplement 1B*). We also tested whether Hsp42 would be upregulated at the protein level because its upregulation was one of the most prominent, based on transcriptomic data for the *pam16-3* mutant. Because of the lack of an antibody against Hsp42, we used a yeast strain with *hsp42* that was tagged with green fluorescent protein (GFP) that allowed us to follow Hsp42 levels by monitoring the GFP signal. We could only test the consequences of impairments in mitochondrial import that were caused by CCCP treatment in this experimental setup. Consistent with the transcriptomic *pam16-3* data, we observed an increase in the abundance of Hsp42 (*Figure 3D*). After identifying Hsp42 and Hsp104 as two molecular chaperones that change significantly when mitochondrial import is impaired, we further examined their expression levels when metastable mitochondrial proteins were overproduced. Hsp104 levels were significantly upregulated upon such protein overproduction (*Figure 3E and F*). Hsp42 levels were again analyzed in the background of the *hsp42-GFP* strain. Here, we also observed an increase in the abundance of Hsp42 for all metastable proteins (*Figure 3G and H*). In the case of pRip1, this response was not observed at the transcriptome level but was significant at the protein level (*Figure 2C* vs. *Figure 3G and H*). Thus, the upregulation of specific molecular chaperones in response to cytosolic mitochondrial protein aggregation could be achieved both transcriptionally and post-transcriptionally. In the case of *pam16-3* mutant, the transcriptomic response involved a number of chaperone genes, whereas on the protein level we observed a significant response of the aggregate-related chaperones, Hsp42 and Hsp104. Interestingly, in the case of pRip1 overproduction, Hsp42 and Hsp104 were increased without any transcriptional stimulation, meaning that in the case of mitochondrial precursor aggregates the post-transcriptional response can be uncoupled from transcription.

In yeast, different types of aggregates have been identified (*Sontag et al., 2017*; *Tyedmers et al., 2010*). Both Hsp42 and Hsp104 are molecular chaperones that are involved in aggregate handling in inclusion bodies (*Balchin et al., 2016*; *Mogk et al., 2015*; *Mogk et al., 2019*). To assess whether Hsp42 and Hsp104 upregulation is associated with a response to protein aggregation, we tested whether they can recognize metastable mitochondrial protein aggregates. We labeled FLAG-tagged

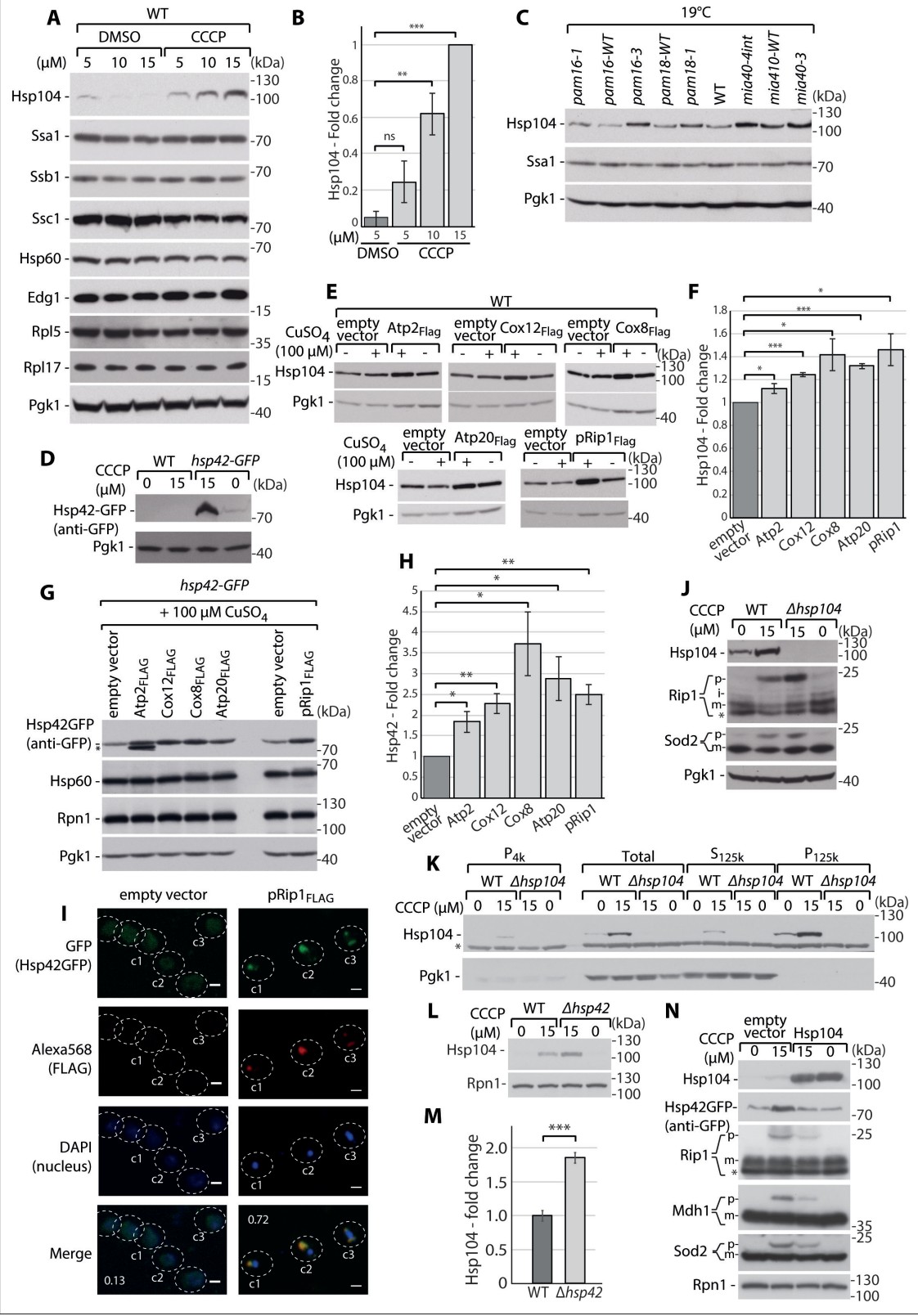

**Figure 3.** Cytosolic mitochondrial protein aggregation elicits a molecular chaperone response to restore cellular homeostasis. ( A-D) Hsp104 and Hsp42 are upregulated in response to impairment in mitochondrial protein import. (**A**) Total protein cell extracts from WT yeast cells were grown at 24 °C and treated with 0, 5, 10, or 15 µM carbonyl cyanide m-chlorophenyl hydrazine (CCCP) for 30 min. (**B**) Quantified changes in Hsp104 protein expression from (**A**). Quantified data are shown as mean ± SEM. n = 3. (**C**) Total protein cell extracts from WT (YPH499), *pam16-WT*, *pam16-1*, *pam16-3*, *pam18-WT*,

*Figure 3 continued on next page*

*Figure 3 continued*

*pam18-1*, *mia40-4WT*, *mia40-4int*, and *mia40-3* grown at a permissive temperature of 19 °C and shifted to a restrictive temperature of 37 °C for 3 hr. (**D**) Total protein cell extracts from WT yeast cells and *hsp42-GFP* yeast grown at 24 °C and treated with 0 or 15 µM CCCP for 30 min. (**E–H**) Metastable protein overexpression increases the expression levels of molecular chaperones. Total protein cell extracts that expressed the indicated metastable proteins or an empty vector control for 4 hr showed higher levels of Hsp104 (**E**) and Hsp42 (**G**). (**F, H**) Quantitative analyses of Hsp104 (**F**) and Hsp42 (**H**) levels from (**E**) and (**G**), respectively, are shown. Quantified data are shown as the mean ± SEM. n = 3. (**I**) Aggregated metastable proteins co-localize with Hsp42-GFP. Representative confocal microscope images show metastable proteins that were tagged with Alexa568 fluorophore in the *hsp42-GFP* yeast strain. Scale bar = 2 µm. See Materials and methods for further details. Pearson's correlation coefficients were calculated for the indicated cells for each condition. (**J**) Total protein cell extracts from WT yeast cells and *Δhsp104* yeast grown at 24 °C and treated with 0 or 15 µM CCCP for 30 min. (**K**) SDS-PAGE analysis of aggregation assay fractions of samples of WT yeast cells and *Δhsp104* yeast grown at 24 °C and treated with 0 or 15 µM CCCP for 30 min, with 2 % sucrose as the carbon source. (**L**) Total protein cell extracts from WT yeast cells and *Δhsp42* yeast grown at 24 °C and treated with 0 or 15 µM CCCP for 30 min. (**M**) Quantified changes in Hsp104 expression from (**L**). Quantified data are shown as the mean ± SEM. n = 3. (**N**) Total protein cell extracts from WT yeast cells and yeast cells that expressed Hsp104 grown overnight at 24 °C and treated with 0 or 15 µM CCCP for 30 min. In (**A**, **C–E**, **G**, **J–L**, **N**), the samples were separated by SDS-PAGE and identified by western blot with specific antisera. Each experiment was repeated three times. p: presequence protein; m: mature protein; *: nonspecific. Significance in (**B**, **F**, **H**, **M**): *p<0.05, **p≤0.01, ***p≤0.001, ****p≤0.0001; ns: nonsignificant.

The online version of this article includes the following figure supplement(s) for figure 3:

**Figure supplement 1.** Total protein cell extract changes that were attributable to mitochondrial import defects.

**Figure supplement 2.** Cellular effects of Hsp42 and Hsp104 overexpression during mitochondrial import defects.

metastable proteins with the Alexa568 fluorophore and followed its co-localization with Hsp42-GFP by confocal microscopy. We observed the strong accumulation of pRip1 in the form of inclusion body-like large deposits in the cytosol (*Figure 3I*). These deposits co-localized with the GFP signals, indicating that Hsp42 sequestered pRip1 aggregates. Next, we reasoned that if these specific molecular chaperones were responsible for preventing the aggregation of metastable proteins, then we should observe a change in the amount of accumulated proteins when they are not present in the cell. In cells with *hsp104* deletion, we observed an increase in the accumulation of pRip1 upon CCCP treatment (*Figure 3J*). Thus, Hsp104 function was essential for limiting aggregates of mitochondrial precursors. Based on the aggregation assay, we detected Hsp104 in the pellet fraction at higher amounts than without CCCP treatment, suggesting that an additional amount of Hsp104 associated with aggregates (*Figure 3K*). The lack of Hsp42 did not result in the upregulation of pRip1, pSod2, or pMdh1 (*Figure 3—figure supplement 2A and B*). However, the *Δhsp42* strain exhibited the significant upregulation of Hsp104, suggesting a compensatory mechanism that diminishes negative consequences of Hsp42 deletion (*Figure 3L and M*). Next, we tested whether the overexpression of Hsp104 and Hsp42 exerts beneficial effects when mitochondrial import defects are stimulated. Hsp104 overproduction resulted in lower levels of pRip1, pSod2, and pMdh1 upon CCCP treatment (*Figure 3N*, *Figure 3—figure supplement 2C*). In this case, we again observed a link between Hsp42 and Hsp104 levels, in which higher levels of Hsp104 were associated with lower levels of Hsp42 in the cell (*Figure 3N*, *Figure 3—figure supplement 2D*). The overexpression of Hsp42 did not have such an effect on pRip1, pSod2, or pMdh1, in which levels of p forms did not change (*Figure 3—figure supplement 2E and F*). Finally, we tested whether higher amounts of Hsp42 or Hsp104 exert positive effects on cells that overproduce metastable proteins. Despite the beneficial effect of Hsp104 on precursor aggregate levels, we did not observe the rescue of growth defects that were observed for strains that expressed Atp2, Cox8, Cox12, pRip1, or Atp20 when either Hsp104 or Hsp42 was overproduced (*Figure 3—figure supplement 2G*). Altogether, our results show that the aggregate-related chaperone response is activated in the cell to handle consequences of cytosolic accumulation of mitochondrial precursor proteins.

## Mitochondrial protein import failure impairs cellular protein homeostasis

We next investigated whether the presence of metastable and aggregation-prone mitochondrial precursors initiates the accumulation of p forms of other mitochondrial proteins. We verified that this was the case for some of the metastable proteins. The overproduction of Atp2 and Cox8 resulted in the concurrent accumulation of other mitochondrial proteins, pRip1 and pSod2, in the total cell fractions, without any other stimulation beyond just the overexpression of metastable proteins (*Figure 4A–C*,

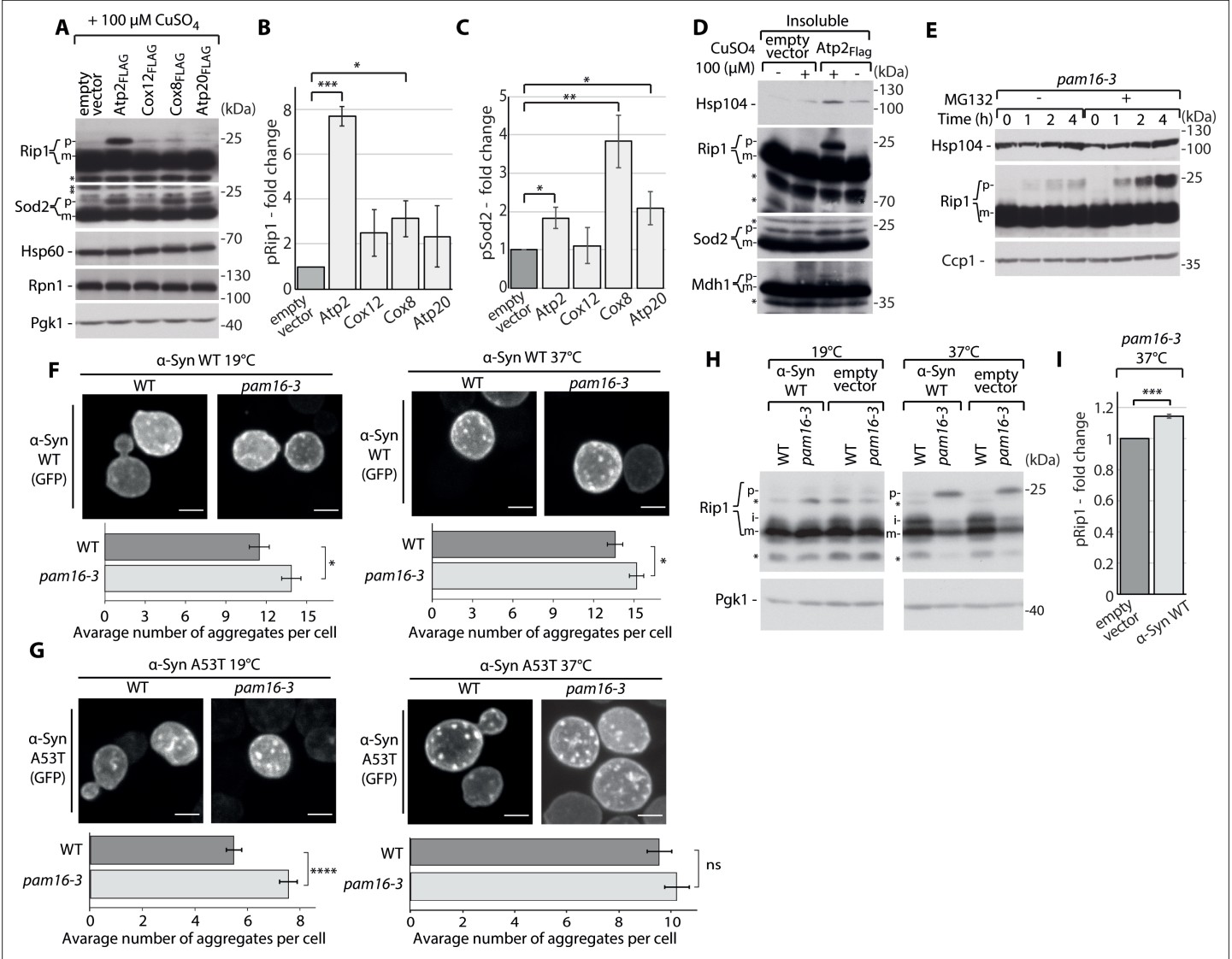

**Figure 4.** Mitochondrial protein import dysfunction enhances impairment in cellular homeostasis. (**A–D**) Metastable proteins cause the accumulation and aggregation of other mitochondrial precursor proteins. (**A**) Total protein cell extracts from *hsp42-GFP* cells that expressed selected metastable proteins or an empty vector control overnight. Changes for pRip1 (**B**) and pSod2 (**C**) were quantified. Quantified data are shown as the mean ± SEM. n = 3. (**D**) SDS-PAGE analysis of aggregation assay fractions of *hsp42-GFP* yeast cells that overexpressed Atp2$_{FLAG}$ or an empty vector control for 3 hr, with 2 % sucrose as the carbon source. Insoluble, S$_{4k}$ aggregation assay fraction. (**E**) Total protein cell extract from *pam16-3* mutant yeast strains treated with 75 μM MG132 for 1 hr under permissive growth conditions and subsequently heat shocked at 37 °C for 0, 1, 2, or 4 hr. (**F, G**) Representative confocal images of α-Syn WT-GFP (**F**) and A53T-GFP (**G**) aggregates in WT (*pam16-WT*) and *pam16-3* yeast strains. α-Syn WT-GFP and A53T-GFP were induced for 4 hr at 19 °C and for an additional 2 hr at 19 °C for control or at 37 °C for heat shock. Scale bar = 2 μm. See Materials and methods for further details. The bar plot shows the average number of aggregates per cell. The data are shown as the mean ± SEM. n = 57–83 for α-Syn WT-GFP. n = 154–175 for α-Syn A53T-GFP. (**H**) Total cell extracts of WT (*pam16-WT*) and *pam16-3* yeast strains that expressed α-Syn WT-GFP induced for 4 hr at 19 °C and for an additional 2 hr at 19 °C for control or 37 °C for heat shock. (**I**) Quantitative analysis of pRip1 from (**H**). Quantified data are shown as the mean ± SEM. n = 3. In (**A, D, E, H**), protein samples were separated by SDS-PAGE and identified by western blot with specific antisera. Each experiment was repeated three times. For western blot: p: presequence protein; i: intermediate protein; m: mature protein; *: nonspecific. *p<0.05, **p≤0.01, ***p≤0.001, ****p≤0.0001; ns: nonsignificant.

The online version of this article includes the following figure supplement(s) for figure 4:

**Source data 1.** Source data for the average number of aggregates per cell and average aggregates size: α-Syn WT-GFP and α-Syn A53T-GFP for WT (*pam16-WT*) and *pam16-3* strains at 19 and 37°C.

**Figure supplement 1.** Effects of mitochondrial protein import dysfunction on cellular homeostasis.

*Figure 4—figure supplement 1A*). Atp20 overproduction also stimulated a significant increase in

pSod2 but not pRip1 in the total cell fractions (*Figure 4A–C*). We hypothesized that differences in pRip1 and pSod2 accumulation for analyzed metastable proteins may reflect the differences: in the mitochondria import rates of these metastable proteins, in their relative cytosolic abundance, and in the protein sequence affecting their aggregation dynamics. Nevertheless, Rip1, Sod2, and Mdh1 co-aggregated with Atp2 and Cox8 metastable proteins in the insoluble fraction, based on the aggregation assay analysis (*Figure 4D*, *Figure 4—figure supplement 1B*). Thus, the greater abundance of aggregation-prone mitochondrial precursors resulted in the progression of mitochondrial protein import defect and consequently the larger cytosolic aggregation of mitochondrial precursors. This observation may justify why Atp2, Cox8, and Atp20 exhibited the most pronounced viability decrease, demonstrated by the drop-test results (*Figure 1B*). Moreover, this process was partially counteracted by proteasome-mediated degradation. MG132-mediated protein inhibition led to an increase in the accumulation and aggregation of mitochondrial precursor proteins (*Figure 4E*, *Figure 4—figure supplement 1C*).

Finally, we tested whether a chain reaction may occur, wherein the cytosolic aggregation of mitochondrial proteins first impaired the import efficiency and solubility of other mitochondrial precursors and subsequently induced the downstream aggregation of other non-mitochondrial proteins. We took advantage of α-synuclein (α-Syn), a model protein of neurodegenerative disorders. α-Syn WT and two of its mutations, A30P and A53T, are implicated in Parkinson's disease. α-Syn WT and A53T, but not A30P, share a similar cellular distribution when they are expressed, which allows monitoring the formation of their aggregates in the cell (*Outeiro and Lindquist, 2003*). Therefore, we used these two model systems that were tagged with GFP in our studies. Confocal microscopy was used to monitor changes in the average number of α-Syn WT and A53T aggregates per cell by following the signal of the GFP tag in response to impairments in mitochondrial import. Here, mitochondrial defects were stimulated by the *pam16-3* mutant. The average number of α-Syn aggregates per cell increased for *pam16-3* at a permissive temperature of 19 °C for both α-Syn WT-GFP and A53T-GFP compared with WT strain (*Figure 4F and G*, *Figure 4—figure supplement 1D and E*, *Figure 4—source data 1a*). The increase in the average number of aggregates continued to grow for α-Syn WT-GFP at the restrictive temperature of 37 °C (*Figure 4F*). The average number of α-Syn A53T-GFP aggregates was higher at 37 °C than at the permissive temperature and did not significantly increase upon the mitochondrial import defect in the *pam16-3* mutant (*Figure 4G*). This observation might be justified by the different morphology of WT and A53 α-Syn aggregates. The A53T mutation of α-Syn resulted in larger aggregates compared with α-Syn WT (*Outeiro and Lindquist, 2003*). To test this hypothesis, we investigated the size of aggregates of α-Syn WT and A53T. We did not observe any difference in the average size of aggregates for α-Syn WT at 19 and 37°C for both WT and *pam16-3* mutant (*Figure 4—figure supplement 1F*, *Figure 4—source data 1*). In the case of α-Syn A53T, puncta were larger and better resolved than α-Syn WT puncta (*Figure 4G*) in both strains. We observed an increase in the average α-Syn A53T aggregate size for WT and the *pam16-3* mutant at 37 °C (*Figure 4—figure supplement 1G*, *Figure 4—source data 1*) when compared to 19 °C. These findings suggest that for α-Syn A53T at 37 °C the average number of aggregates did not significantly change after reaching a threshold and instead they were becoming larger as the aggregation progressed. Finally, we observed the higher accumulation of pRip at combined conditions of a *pam16-3*-stimulated mitochondrial import defect and α-Syn aggregation at 37 °C (*Figure 4H and I*, *Figure 4—figure supplement 1D and E*). Thus, the protein aggregation linked to mitochondrial import defects had a dual effect: (i) α-Syn aggregation was stimulated and (ii) α-Syn aggregation further deepened the aggregation of mitochondrial precursor proteins.

Our results indicated that the mitochondrial precursor aggregation in the cytosol led to an increase in the aggregation of other non-mitochondrial proteins, and this acceleration of protein aggregation, which may be compared to a snowball effect, reduced cellular protein homeostasis.

## Mitochondrial dysfunction results in protein aggregation in *Caenorhabditis elegans*

To assess whether mitochondrial precursor aggregation that is caused by mitochondrial protein import deficiency compromises protein homeostasis at the organismal level, we used *C. elegans* as a model system. We used an RNA interference (RNAi) approach to silence *dnj-21* (a homolog of yeast Pam18) in *C. elegans*. The depletion of DNJ-21 stimulates the mitochondrial import defect similarly

to the defect that is observed in yeast for *pam16-3*. We first monitored the aggregation of two model proteins in cytosol, red fluorescent protein (RFP) and GFP, in the transgenic strain that expressed RFP and GFP in the body wall muscle to assess whether the RNAi silencing of *dnj-21* in early adulthood in *C. elegans* is sufficient to stimulate their aggregation (*Figure 5A–C*, *Figure 5—figure supplement 1A*, *Figure 5—source data 1*). We found that the RNAi of *dnj-21* increased the cytosolic aggregation of both GFP and RFP model proteins. Therefore, we next tested whether changes in cellular homeostasis that are attributable to mitochondrial defects affect the health of *C. elegans* when α-Syn is produced. The silencing of *dnj-21*, accompanied by α-Syn expression, decreased worm fitness, manifested by a slower speed and fewer bends compared with the effect of *dnj-21* silencing alone (*Figure 5D*, *Figure 5—source data 2*). Finally, we investigated whether the link between mitochondrial dysfunction is only limited to α-Syn or whether similar effects are observed for other proteins that are linked to neurodegenerative diseases. For this reason, we analyzed if mitochondrial dysfunction enhances amyloid β (Aβ) aggregation in *C. elegans*. We used worms that carried Aβ peptides in body wall muscles and exhibited paralysis in adults when the temperature shifted to 25 °C (*Figure 5—figure supplement 1B*; *Sorrentino et al., 2017*). The level of Aβ aggregates increased at 22 °C when *dnj-21* was silenced (*Figure 5E and F*, *Figure 5—figure supplement 1C and D*, *Figure 5—source data 2*). This stimulated aggregation resulted from the mitochondrial import defect without any accompanying changes in the expression of Aβ peptide (*Figure 5—figure supplement 1E-G* , *Figure 5—source data 2*). We found that Aβ aggregation was accompanied by a decrease in worm motility with *dnj-21* silencing (*Figure 5G*, *Figure 5—figure supplement 1H*, *Figure 5—source data 2*). Overall, our results indicate that the mechanism of aggregation stimulation that is caused by mitochondrial dysfunction is conserved between species, as demonstrated for both α-Syn and Aβ aggregates.

## Discussion

The present study showed that a group of mitochondrial proteins that are downregulated in Alzheimer's disease (i.e., Rip1, Atp2, Cox8, and Atp20) can aggregate in the cytosol and that the overexpression of these proteins upregulates Hsp42 and Hsp104, two molecular chaperones that are associated with inclusion bodies (*Balchin et al., 2016*; *Mogk et al., 2015*; *Mogk et al., 2019*). Depending on how mitochondrial precursor protein aggregation is stimulated, the Hsp42 and Hsp104 upregulation is modulated transcriptionally and post-transcriptionally, with a possibility of post-transcriptional stimulation being a dominant component. Thus, our findings of modulated aggregate-related chaperones upregulation further contribute to the transcriptomic response mechanisms involving chaperones due to mitochondrial import machinery overload (*Boos et al., 2019*), and mechanism of proteasomal protein degradation machinery upregulation (*Wrobel et al., 2015*). We show that stress responses that are induced by mitochondrial proteins, such as the response that was identified herein, mitigate the danger that is related to the aberrant formation of aggregates by metastable mitochondrial precursor proteins.

Our study demonstrated that mitochondrial import defects, triggered either chemically or by mutations of the TIM23 translocase, resulted in the accumulation and aggregation of mitochondrial precursor proteins. Consistent with the experiments that utilized the overproduction of metastable mitochondrial proteins, an Hsp42- and Hsp104-specific molecular chaperone response was also triggered in the cell with mitochondrial import defects. Additionally, the ubiquitin-proteasome system could mitigate this process to some extent (*Wrobel et al., 2015*).

The unresolved aggregation of mistargeted metastable mitochondrial precursor proteins eventually resulted in a chain reaction that led to the accumulation of deposits that were formed by other non-mitochondrial proteins. The collapse of cellular homeostasis resulted in an increase in pRip1 and α-Syn aggregation when mitochondrial import was impaired. More generally, we found that an increase in the aggregation of proteins linked to age-related degenerations under conditions of mitochondrial protein import dysfunction was a conserved cellular mechanism that was observed in both yeast and *C. elegans*. We found that the aggregation of α-Syn and Aβ increased because of mitochondrial dysfunction at the organismal level, accompanied by a decrease in worm fitness.

Several stress response pathways have recently been identified that counteract defects of mitochondrial protein import (*Boos et al., 2019*; *Izawa et al., 2017*; *Kim et al., 2016*; *Martensson et al., 2019*; *Priesnitz and Becker, 2018*; *Wang and Chen, 2015*; *Weidberg and Amon, 2018*; *Wrobel et al., 2015*; *Wu et al., 2019*; *Poveda-Huertes et al., 2020*). Unknown, however, is whether they act

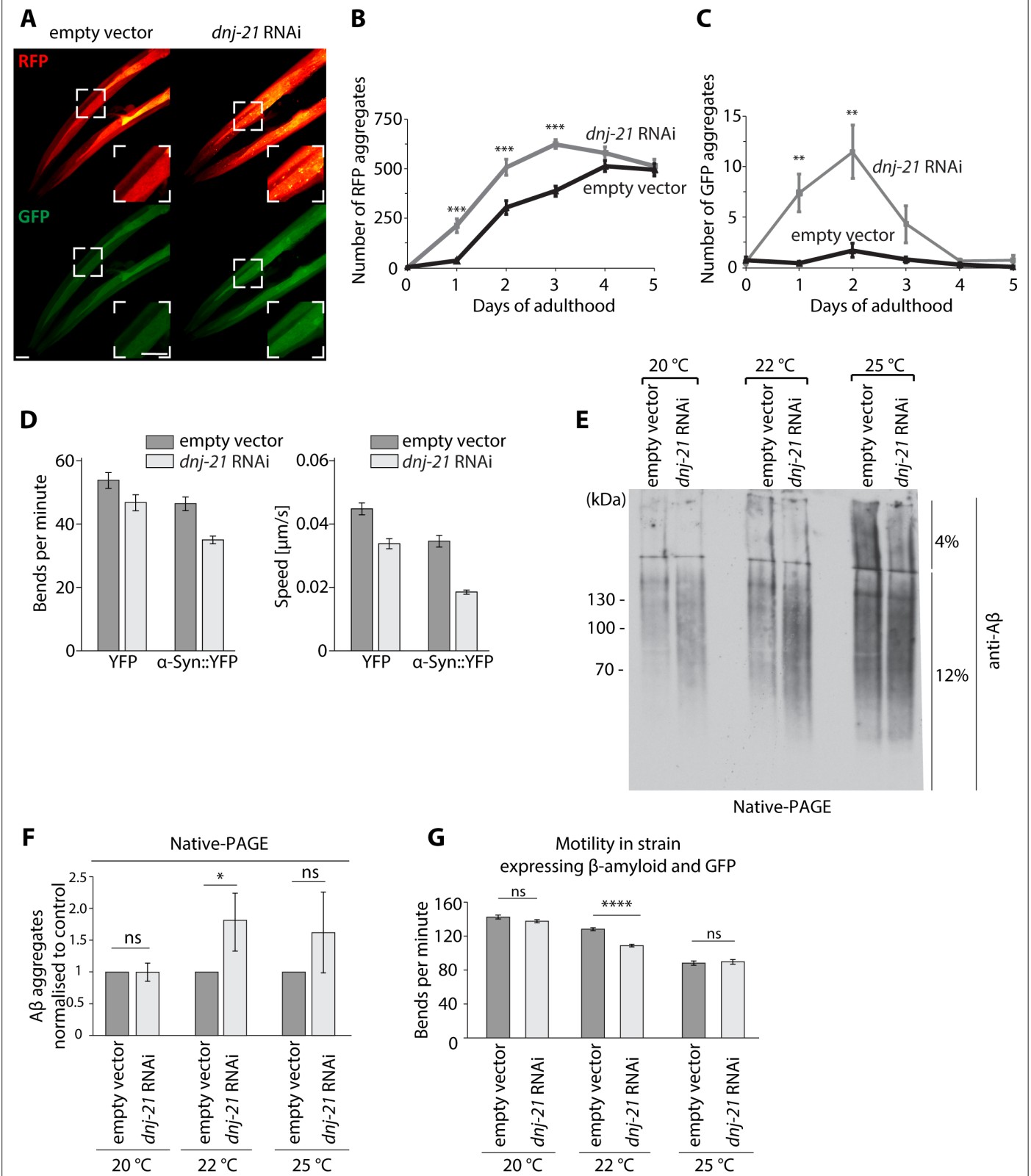

**Figure 5.** Mitochondrial dysfunction results in the accumulation of Aβ aggregates in *C. elegans*. (**A–C**) Mitochondrial dysfunction stimulates the aggregation of model proteins in *C. elegans*. (**A**) Confocal images of worms that expressed wrmScarlet and green fluorescent protein (GFP) in body wall muscle [pmyo-3::wrmScarlet+ pmyo::GFP]. The zoomed image is presented in the white box. Scale bar = 20 μm. (**B**) Number of red fluorescent protein (RFP) aggregates at different days of adulthood of strain [pmyo-3::wrmScarlet+ pmyo-3::GFP] strain upon *dnj-21* RNAi. n = 14–16 worms for

*Figure 5 continued on next page*

*Figure 5 continued*

empty vector. n = 8–16 worms for *dnj-21* RNAi. (**C**) Number of GFP aggregates present at different days of adulthood of [pmyo-3::wrmScarlet+ pmyo-3::GFP] strain upon *dnj-21* RNAi. n = 14–16 worms for empty vector. n = 8–16 worms for *dnj-21* RNAi. (**D**) Motility of Parkinson's disease model strain that expressed α-Syn::YFP in the body wall muscle or control strain that expressed YFP in the body wall muscle upon the silencing of *dnj-21*. An empty vector was used as the control. Data were obtained using an automated body bend assay. The data are shown as the mean ± SEM, with at least n = 700 for each condition. (**E**) Protein aggregation under native conditions in worms that expressed Aβ upon *dnj-21* RNAi. Worms were cultured at 20 °C or shifted to 22 or 25 °C. n = 3. (**F**) Aβ levels were calculated from the native aggregation data in (**E**). The data are shown as the mean ± SD. n = 3. (**G**) Motility in worms that expressed Aβ and GFP upon *dnj-21* RNAi. The data are shown as the mean ± SEM. n = 50 worms per condition. Overall differences between conditions were assessed by unpaired *t*-test by assuming equal variance. *p<0.05, **p≤0.01, ***p≤0.001, ****p≤0.0001; ns: nonsignificant.

The online version of this article includes the following figure supplement(s) for figure 5:

**Source data 1.** Source data for worms that expressed wrmScarlet and green fluorescent protein (GFP) in body wall muscle.

**Source data 2.** Source data for worms that expressed α-Syn::YFP and Aβ.

**Figure supplement 1.** Scheme and controls for experiments on mitochondrial dysfunction in *C. elegans* that resulted in Aβ accumulation.

independently or whether concurrent actions of all of them are required to secure balanced cellular protein homeostasis. The cytosolic responses, which are aiming to clear clogged translocase of the outer membrane (TOM) and the precursor proteins prior to their import through proteasomal activity, are accompanied by the aggregate-specific molecular chaperone response identified in the present study. In addition, recent findings suggest that some precursor proteins can accumulate within mitochondria to form insoluble aggregates (*Poveda-Huertes et al., 2020*). We postulate that when these defense mechanisms may not be sufficient to prevent mitochondrial precursor proteins aggregation in the cytosol, this aggregation will lead to the snowball effect, where aggregation of mitochondrial proteins stimulates further cytosolic aggregation of other proteins. These findings also indicate that precursor protein-induced stress responses are orchestrated to complement each other (*Boos et al., 2020*; *Mohanraj et al., 2020*).

In conclusion, our findings suggest a model in which a cascade of events that is triggered by the cytosolic aggregation of specific metastable mitochondrial proteins contributes to the collapse of cellular protein homeostasis accelerating the aggregation of other proteins, including some that are hallmarks of age-related degeneration. Consequently, defects in mitochondrial function, which are commonly observed during aging and in neurodegeneration, trigger a vicious cycle of protein aggregation. This notion is essential for understanding cellular events that contribute to the onset and progression of neurodegenerative processes. Overall, our results illustrate an important aspect of the interdependence of mitochondrial fitness and cellular protein homeostasis systems, with direct relevance to common neurodegenerative disorders and mitochondrial function. Our findings may suggest new avenues for therapeutic interventions to cure or prevent diseases that are linked to protein aggregation and mitochondrial dysfunction.

# Materials and methods

**Key resources table**

| Reagent type (species) or resource | Designation | Source or reference | Identifiers | Additional information |
|---|---|---|---|---|
| Strain, strain background (*Saccharomyces cerevisiae*) | YPH499 | Standard laboratory strain | In-house ID: 524 | MATa, ade2-101, his3-D200, leu2-D1, ura3-52, trp1-D63, lys2-801 |
| Strain, strain background (*Saccharomyces cerevisiae*) | BY4741 | Standard laboratory strain | In-house ID: 755 | MATa, his3D1, leu2D0, met15D0, ura3D0 |
| Strain, strain background (*Saccharomyces cerevisiae*) | *Δhsp104* | Euroscarf | In-house ID: 1001 | |
| Strain, strain background (*Saccharomyces cerevisiae*) | *hsp42-GFP* | Invitrogen | In-house ID: 1005 | |
| Strain, strain background (*Saccharomyces cerevisiae*) | *Hsp42-BY4741* | Provided by Dr. Bernd Bukau | In-house ID: 1178 | -His selection; Gal1 promoter |

*Continued on next page*

*Continued*

| Reagent type (species) or resource | Designation | Source or reference | Identifiers | Additional information |
|---|---|---|---|---|
| Strain, strain background (*Saccharomyces cerevisiae*) | *pam16WT* | 10.1038/nsmb735 | In-house ID: 736 | |
| Strain, strain background (*Saccharomyces cerevisiae*) | *pam16-1* | 10.1038/nsmb735 | In-house ID: 733 | |
| Strain, strain background (*Saccharomyces cerevisiae*) | *pam16-3* | 10.1038/nsmb735 | In-house ID: 734 | |
| Strain, strain background (*Saccharomyces cerevisiae*) | *pam18WT* | 10.1083/jcb.200308004 | In-house ID: 738 | |
| Strain, strain background (*Saccharomyces cerevisiae*) | *pam18-1* | 10.1083/jcb.200308004 | In-house ID: 739 | |
| Strain, strain background (*Saccharomyces cerevisiae*) | *mia40WT* | 10.1038/sj.emboj.7600389 | In-house ID: 398 | |
| Strain, strain background (*Saccharomyces cerevisiae*) | *mia40-3* | 10.1038/sj.emboj.7600389 | In-house ID: 178 | |
| Strain, strain background (*Saccharomyces cerevisiae*) | *mia40-4int* | 10.1083/jcb.200804095 | In-house ID: 739 | |
| Strain, strain background (*Caenorhabditis elegans*) | Worms expressing GFP | *Caenorhabditis* Genetics Center (CGC) | RRID:WB-STRAIN:WBStrain00005101 CL2122 dvIs15 [(pPD30.38) unc-54(vector) + (pCL26) mtl-2::GFP] | Control strain for GM101 |
| Strain, strain background (*Caenorhabditis elegans*) | Worms expressing Aβ and GFP | *Caenorhabditis* Genetics Center (CGC) | RRID:WB-STRAIN:WBStrain00007866 GMC101 dvIs100 [punc-54::A-beta-1–42::unc-54–3'-UTR+ mtl-2p::GFP] | |
| Strain, strain background (*Caenorhabditis elegans*) | α-syn::YFP | 10.1371/journal.pgen.1000027 | OW40 zgIs15 [punc-54::αsyn::YFP] IV | |
| Strain, strain background (*Caenorhabditis elegans*) | YFP | 10.1371/journal.pgen.1000027 | OW450 rmIs126 [punc-54::YFP]V | Control strain for OW40 |
| Strain, strain background (*Caenorhabditis elegans*) | [pmyo-3::wrmScarlet+ pmyo::GFP] | A strain generated in our laboratory | ACH87: wacIs11 [pmyo-3::mGFP::SL2 gdp-2-wrmScarlet::unc-54-3´UTR, unc-119(+)] | |
| Strain, strain background (*Escherichia coli*) | HB101 | *Caenorhabditis* Genetics Center (CGC) | RRID:WB-STRAIN:WBStrain00041075 HB101 | *E. coli* strain used as a food source for worms |
| Strain, strain background (*Escherichia coli*) | HT115(DE3) | *Caenorhabditis* Genetics Center (CGC) | RRID:WB-STRAIN:WBStrain00041074 HT115(DE3) | *E. coli* strain used as a food source for worms in RNAi experiments |
| Strain, strain background (*Escherichia coli*) | OP50 | *Caenorhabditis* Genetics Center (CGC) | RRID:WB-STRAIN:WBStrain00041971 OP50 | *E. coli* strain used as a food source for worms |
| Antibody | Anti-Rip1 (rabbit polyclonal) | Custom made | | WB (1:500) |
| Antibody | Anti-Sod2 (rabbit polyclonal) | Custom made | | WB (1:500) |
| Antibody | Anti-Mdh1 (rabbit polyclonal) | Custom made | | WB (1:500) |
| Antibody | Anti-Hsp60 (rabbit polyclonal) | Custom made | | WB (1:500) |
| Antibody | Anti-Pgk1 (rabbit polyclonal) | Custom made | | WB (1:500) |
| Antibody | Anti-Tom70 (rabbit polyclonal) | Custom made | | WB (1:500) |

*Continued on next page*

*Continued*

| Reagent type (species) or resource | Designation | Source or reference | Identifiers | Additional information |
|---|---|---|---|---|
| Antibody | Anti-Ssa1 (rabbit polyclonal) | Custom made | | WB (1:500) |
| Antibody | Anti-Ssb1 (rabbit polyclonal) | Custom made | | WB (1:500) |
| Antibody | Anti-Ssc1 (rabbit polyclonal) | Custom made | | WB (1:500) |
| Antibody | Anti-Rpl17 (rabbit polyclonal) | Custom made | | WB (1:500) |
| Antibody | Anti-Ccp1 (rabbit polyclonal) | Custom made | | WB (1:500) |
| Antibody | Anti-Qcr8 (rabbit polyclonal) | Custom made | | WB (1:500) |
| Antibody | Anti-Qcr6 (rabbit polyclonal) | Custom made | | WB (1:500) |
| Antibody | Anti-Cox12 (rabbit polyclonal) | Custom made | | WB (1:500) |
| Antibody | Anti-Tom20 (rabbit polyclonal) | Custom made | | WB (1:500) |
| Antibody | Anti-Edg1 (rabbit polyclonal) | Custom made | | WB (1:500) |
| Antibody | Anti-Cdc48 (rabbit polyclonal) | Custom made | | WB (1:500) |
| Antibody | Anti-Rpl5 (rabbit polyclonal) | Custom made | | WB (1:500) |
| Antibody | Anti-Dnj21 (rabbit polyclonal) | Custom made | | WB (1:500) |
| Antibody | Anti-Rpn1 (rabbit polyclonal) | Custom made | | WB (1:500) |
| Antibody | Anti-Hsp42 (rabbit polyclonal) | Provided by Dr. Bernd Bukau | | WB (1:5000) |
| Antibody | Anti-Hsp104 (rabbit polyclonal) | Enzo | RRID:AB_11181448 ADI-SPA-1040-F | WB (1:1000) |
| Antibody | Anti-GFP (mouse monoclonal) | Sigma | RRID:AB_390913 11814460001 | WB (1:1000) |
| Antibody | anti-3-PGDH (rabbit polyclonal) | Millipore | RRID:AB_2783876ABS571 | IF(1:1000), WB (1:1000) |
| Antibody | Anti-beta amyloid 1–16 (mouse monoclonal) | BioLegend | RRID:AB_2564653 803001 | WB (1:1000) |
| Antibody | Anti-Tubulin (mouse monoclonal) | Sigma | RRID:AB_477593 T9026 | WB (1:500) |
| Antibody | Anti-mCherry (anti-wrmScarlet/RFP); (rabbit polyclonal) | Abcam | RRID:AB_2571870 ab167453 | WB (1:1000) |
| Antibody | Anti-FLAG (mouse monoclonal) | Sigma | RRID:AB_262044 F1804 | IF (1:1000) WB (1:1000) |
| Antibody | Alexa Fluor 568 (goat anti-mouse) | Invitrogen | RRID:AB_144696 A11031 | IF (1:500) |
| Recombinant DNA reagent | L4440 (control for RNAi in *C. elegans*) (plasmid) | AddGene | #1654 | |
| Recombinant DNA reagent | *dnj-21* RNAi (plasmid) | Recombinant DNA generated in our laboratory | pMJS1 In-house ID: 435p | Plasmid used for silencing of *dnj-21* |
| Recombinant DNA reagent | Hsp104-pFL46L (plasmid) | 10.1128/MCB.20.19.7220–7229.2000 | | Provided by Dr. Magdalena Boguta |

*Continued on next page*

*Continued*

| Reagent type (species) or resource | Designation | Source or reference | Identifiers | Additional information |
|---|---|---|---|---|
| Recombinant DNA reagent | α-Syn WT-GFP (plasmid) | 10.1126/science.1090439 | | Provided by Dr. Tiago Outeiro |
| Recombinant DNA reagent | α-Syn A53T-GFP (plasmid) | 10.1126/science.1090439 | | Provided by Dr. Tiago Outeiro |
| Recombinant DNA reagent | Empty vector with FLAG-tag, pCup1 (plasmid) | Recombinant DNA generated in our laboratory | pPCh26 In-house ID: 435p | See *Supplementary file 3* |
| Recombinant DNA reagent | Cox8 (plasmid) | Recombinant DNA generated in our laboratory | pPCh17 In-house ID: 481p | See *Supplementary file 3* |
| Recombinant DNA reagent | Atp2 (plasmid) | Recombinant DNA generated in our laboratory | pPCh18 In-house ID: 482p | See *Supplementary file 3* |
| Recombinant DNA reagent | Cor1 (plasmid) | Recombinant DNA generated in our laboratory | pPCh19 In-house ID: 484p | See *Supplementary file 3* |
| Recombinant DNA reagent | Mas1 (plasmid) | Recombinant DNA generated in our laboratory | pPCh20 In-house ID: 485p | See *Supplementary file 3* |
| Recombinant DNA reagent | Qcr6 (plasmid) | Recombinant DNA generated in our laboratory | pPCh21 In-house ID: 486p | See *Supplementary file 3* |
| Recombinant DNA reagent | Qcr8 (plasmid) | Recombinant DNA generated in our laboratory | pPCh22 In-house ID: 487p | See *Supplementary file 3* |
| Recombinant DNA reagent | pRip1 (plasmid) | Recombinant DNA generated in our laboratory | pPCh23 In-house ID: 488p | See *Supplementary file 3* |
| Recombinant DNA reagent | iRip1 (plasmid) | RDNA generated in our laboratory | pPCh24 In-house ID: 489p | See *Supplementary file 3* |
| Recombinant DNA reagent | mRip1 (plasmid) | Recombinant DNA generated in our laboratory | pPCh29 In-house ID: 490p | See *Supplementary file 3* |
| Recombinant DNA reagent | Atp20 (plasmid) | Recombinant DNA generated in our laboratory | pPCh30 In-house ID: 483p | See *Supplementary file 3* |
| Recombinant DNA reagent | Cox12 (plasmid) | Recombinant DNA generated in our laboratory 10.1186/s12915-018-0536-1 | pPB36.1 In-house ID: 471p | See *Supplementary file 3* |
| Sequence-based reagent | Primers | This paper | PCR primers | See *Supplementary file 2* |
| Commercial assay or kit | RNeasy Mini Kit | Qiagen | 74104 | |
| Chemical compound, drug | CCCP | Sigma | C2759 | |
| Chemical compound, drug | MG132 | Enzo | BML-PI102-0005 | |
| Software, algorithm | ZEN | Zeiss | ZEN 2012 SP5 FP3 (black) | Confocal Microscopy Data Collection Software |
| Software, algorithm | FastQC | www.bioinformatics.babraham.ac.uk/projects/fastqc | RRID:SCR_014583 | |
| Software, algorithm | Salmon (v0.11.2) | 10.1038/nmeth.4197 | RRID:SCR_017036 | |

*Continued on next page*

*Continued*

| Reagent type (species) or resource | Designation | Source or reference | Identifiers | Additional information |
|---|---|---|---|---|
| Software, algorithm | DESeq2 version 1.26.0 | 10.1186/s13059-014-0550-8 | RRID:SCR_015687 | |
| Other | ProLong Diamond antifade mountant with DAPI | Thermo Fisher Scientific | P36962 | |

## Experimental design

No statistical methods were used to predetermine sample size. The experiments were not randomized. The investigators were not blinded to allocation during the experiments and outcome assessment. All of the experiments were repeated at least three times. For experiments with a larger number of biological replicates, the *n* is indicated.

## Yeast strains

The *Saccharomyces cerevisiae* strains were derivatives of YPH499 (MATa, ade2-101, his3-D200, leu2-D1, ura3-52, trp1-D63, lys2-801) (524) or BY4741 (MATa, his3D1, leu2D0, met15D0, ura3D0) (755). The descriptions of WT yeast cells refer to the BY4741 strain unless otherwise indicated. For the inducible expression of FLAG-tagged metastable proteins, the amino acid sequences were amplified by polymerase chain reaction (PCR) from yeast genomic DNA. The resulting DNA fragments were cloned in frame with the FLAG tag using the oligonucleotides that are indicated in *Supplementary file 2* into the pESC-URA vector (Agilent), in which the GAL10 and GAL1 promoters were replaced with the Cup1 promoter. This procedure yielded the pPCh17 (481p), pPCh18 (482p), pPCh19 (484p), pPCh20 (485p), pPCh21 (486p), pPCh22 (487p), pPCh23 (488p), pPCh24 (489 p), pPCh26 (492p), pPCh29 (490p), pPCh30 (483p), and pPB36.1 (471p) plasmids (*Kowalski et al., 2018*), with a FLAG tag at the C-terminus and expressed under control of the Cup1 promoter (*Supplementary file 3*). The plasmid that expressed Hsp104 under the endogenous promoter in pFL46L vector was provided by Prof. Magdalena Boguta (IBB PAN) (*Chacinska et al., 2000*). Yeast cells were transformed according to the standard procedure. The *Δhsp104* (1001) deletion yeast strain was purchased from Euroscarf. The *hsp42-GFP* (1005) yeast strain was purchased from Invitrogen. The Hsp42-expressing strain on the BY4741 (1178) background under the Gal1 promoter was provided by Dr. Bernd Bukau laboratory (ZMBH Heidelberg). The temperature-sensitive *pam16-1* (YPH-BG-mia1-1) (733) and *pam16-3* (YPH-BGmia1-3) (734) and corresponding *pam16-WT* (736) were described previously (*Frazier et al., 2004*). The temperature-sensitive *pam18-1* (YPH-BG-Mdj3-66) (739) and its respective *pam18-WT* (738) were described previously (*Truscott et al., 2003*), similar to *mia40-3* (YPH-BGfomp2-8) (178) and *mia40-4int* (305) and the corresponding *mia40-WT* (398) (*Chacinska et al., 2004*; *Stojanovski et al., 2008*).

## Yeast growth, treatments, and analysis

In the experiments in which metastable proteins were overproduced, the strains were grown on minimal synthetic medium (0.67 % [w/v] yeast nitrogen base, 0.079 % [w/v] complete supplement mixture [CSM] amino acid mix without uracil, containing 2 % [w/v] galactose with 0.1 % glucose or 2 % [w/v] sucrose as carbon source). The yeast culture was performed at 28 °C to the early logarithmic growth phase and further induced with 100 μM CuSO$_4$ for 4 hr unless otherwise indicated. For the CCCP (Sigma, catalog no. C2759) treatment experiments, yeast strains were grown in full medium that contained 2 % (w/v) glucose or 2 % (w/v) sucrose at 24 °C to the early logarithmic growth phase, at which time they were treated with 0, 5, 10, or 15 μM CCCP for 15 or 30 min at the growth temperature. The temperature-sensitive mutants were grown at a permissive temperature of 19 °C and analyzed with and without a shift to a restrictive temperature of 37 °C for the indicated time. The temperature-sensitive *pam16-3* mutants, which were transformed with p426 α-Syn WT-GFP or A53T-GFP plasmids with galactose induction, were grown at a permissive temperature of 19 °C to the early logarithmic growth phase. Next, they were further induced with 0.5 % (w/v) galactose for another 4 hr. After induction, the samples were shifted to a restrictive temperature of 37 °C for heat shock or left at 19 °C as a control for 2 hr. p426 GAL α-Syn WT-GFP and A53T-GFP were provided by Dr. Tiago Outeiro (Göttingen). For the experiments that performed proteasome inhibition, the strains were grown on minimal synthetic medium (0.67 % [w/v] yeast nitrogen base, 0.079 % [w/v] CSM amino

acid mix without ammonium sulfate, supplemented with 0.1 % [w/v] proline, 0.03 % [w/v] SDS, and 2 % [w/v] galactose as carbon source) at a permissive temperature of 19 °C to the early logarithmic growth phase. Afterward, 75 µM MG132 (Enzo, catalog no. BML-PI102-0005) was added to the cell culture and maintained at 19 °C or transferred to a restrictive temperature of 37 °C for the indicated time. Proteins were separated by sodium dodecyl sulfate-polyacrylamide gel electrophoresis (SDS-PAGE) using 12 or 15 % gels and then transferred to polyvinyl difluoride (PVDF) membranes. Immunodetection was performed according to standard techniques using chemiluminescence. The following antibodies were used in the study: FLAG (Sigma, catalog no. F1804), GFP (Sigma, catalog no. 11814460001), Hsp104 (Enzo, catalog no. ADI-SPA-1040-F), Alexa Fluor 568 (Invitrogen, catalog no. A11031), Rip1, Sod2, Mdh1, Hsp60, Pgk1, Tom70, Ssa1, Ssc1, Rpl17, Ccp1, Qcr8, Qcr6, Cox12, Tom20, Ssb1, Edg1, Cdc48, Rpn1, Rpl5, and DNJ-21 (these latter antibodies were non-commercial antibodies that are available in our laboratory depository). Hsp42 antibody was provided by Dr. Bernd Bukau laboratory (ZMBH Heidelberg).

## Aggregation assay

For the aggregation assay of metastable proteins, galactose with 0.1 % glucose was used as the carbon source. For the experiments that were followed by MG132 treatment, 2 % galactose was used as the carbon source. For all other experiments, 2 % (w/v) sucrose was used as the carbon source. Ten $OD_{600}$ units of cells were harvested by centrifugation at 5000× $g$ for 5 min at 4 °C and resuspended in 400 µl of lysis buffer (30 mM Tris-HCl pH 7.4, 20 mM KCl, 150 mM NaCl, 5 mM ethylenediaminetetraacetic acid [EDTA], and 0.5 mM phenylmethylsulfonyl fluoride [PMSF]). Cells were homogenized by vortexing with 200 µl of glass beads (425–600 µm, Sigma-Aldrich) using a Cell Disruptor Genie (Scientific Industries) for 10 min at 4 °C. The cell lysate was solubilized with 1 % (v/v) Triton X-100 and mixed gently for 20 min at 4 °C. The sample was centrifuged at 4000× $g$ for 5 min at 4 °C to remove unbroken cells and detergent-resistant aggregates ($P_{4k}$). Next, the supernatant was transferred to two tubes at equal volumes. One was saved as the total protein fraction (T). The second was centrifuged at 125,000× $g$ for 60 min at 4 °C to separate soluble proteins ($S_{125k}$) from protein aggregates ($P_{125k}$). The total (T) and soluble ($S_{125k}$) fractions were precipitated by 10 % trichloroacetic acid. After 30 min of incubation on ice, the samples were centrifuged at 20,000× $g$ for 15 min at 4 °C, washed with ice-cold acetone, and centrifuged again. The pellets from the T and $S_{125k}$ samples, as well as $P_{4k}$ and $P_{125k}$, were resuspended in urea sample buffer (6 M urea, 6 % [w/v] SDS, 125 Tris-HCl pH 6.8, and 0.01 % [w/v] bromophenol blue), incubated for 15 min at 37 °C, and analyzed by SDS-PAGE followed by western blot.

## RNA-seq: sample preparation

Total RNA was isolated from the *pam16-3* mutant and corresponding WT samples, as well as BY4741 cells that were transformed with the pESC-URA plasmid that expressed pRip1-FLAG protein or an empty vector under control of the Cup1 promoter. The empty vector and pRip1 samples were grown at 28 °C to the early logarithmic growth phase and further induced with 100 µM $CuSO_4$ for 1, 4, or 8 hr. The *pam16WT* and *pam16-3* mutant samples after growth to the early logarithmic phase were shifted to a restrictive temperature of 37 °C for 0, 1, 4, or 8 hr. A total of 2.5 $OD_{600}$ units of cells were harvested by centrifugation at 5000× $g$ for 5 min at 4 °C, resuspended in 125 µl of RNAlater solution (Thermo Fisher Scientific, catalog no. AM7020), and incubated for 1 hr at 4 °C. After re-centrifugation, the cell pellet was frozen in liquid nitrogen and stored for further processing.

The *pam16-3-* and pRip1-related samples were prepared for RNA sequencing, including four biological replicates. Each replicate was individually generated from frozen stocks. The defrosted samples were centrifuged at 5000× $g$ for 5 min at 4 °C to thoroughly remove the supernatants. The loosened pellets were resuspended in 200 µl of phosphate-buffered saline (PBS). To this sample was added 600 µl of Buffer RLT (Qiagen, catalog no. 79216) along with acid-washed glass beads, and the cells were shaken in a cell disruptor for 10 min at maximum speed at 4 °C. Further processing was performed with agreement of the RNA extraction protocol using the Qiagen Purification of Total RNA Kit for Yeast with Optimal On-Column DNase Digestion (RNeasy Mini Kit 50; Qiagen, catalog no. 74104).

## RNA-seq: analysis

Sequencing was performed using an Illumina NextSeq500 instrument with v2 chemistry, resulting in an average of 15–20 million reads per library with a 75 bp single-end setup. The resulting raw reads were assessed for quality, adapter content, and duplication rates with FastQC (*Andrews, 2010*). Reads for each sample were aligned to Ensembl *Saccharomyces cerevisiae* transcriptome version 91 (Saccharomyces_cerevisiae.R64-1-1.91.gtf) and quantified using Salmon (v0.11.2) (*Patro et al., 2017*) with default parameters. Full lists of genes for the *pam16-3* and pRip1 samples are available in *Figure 2—source data 1* and *Figure 2—source data 2*, respectively. Expression matrices for the *pam16-3* and pRip1 strains are available in *Figure 2—source data 3* and *Figure 2—source data 4*, respectively. For each time point, differentially expressed genes were identified using DESeq2 version 1.26.0 (*Love et al., 2014*). Only genes with a minimum fold change of $\log2 \pm 1$, maximum Benjamini–Hochberg corrected p value of 0.05, and minimum combined mean of 10 reads were deemed to be significantly differentially expressed. The WT samples were used as a reference in *pam16-3*, and samples with the empty pESC-URA plasmid were the control for the pRip1 comparisons. Raw data were deposited in the GEO repository (accession no. GSE147284). Raw data are available at https://www.ncbi.nlm.nih.gov/geo/query/acc.cgi?acc=GSE147284 using the token mvwtuqeexrgvbut.

Identification of the molecular chaperone and mitoCPR-related genes was based on an unbiased data analysis apart from the KEGG enrichment analysis. Among all significantly changed genes (5 % FDR), we selected genes that encoded non-mitochondrial proteins. From this group, we selected genes for which we could identify at least 10 genes that shared similar function. This allowed for identification of group of proteins that were changed but not enriched in KEGG enrichment analysis.

## KEGG enrichment analysis

The KEGG enrichment analysis was performed for RNA-seq using in-house developed methods based on KEGG.db (v. 3.2.3) and org.SC.sgd.db (v 3.10.0). Pathways and genes that were selected in each developmental stage were filtered after Benjamini–Hochberg correction for an adjusted $p<0.05$.

## Microscopy

Yeast confocal microscopy images of GAL α-Syn WT-GFP and A53T-GFP in the *pam16-3* mutant and its corresponding WT were acquired with a Zeiss LSM700 laser-scanning confocal microscope using a 60× oil objective (NA 1.4). Images were recorded with pixel dimensions of 25 nm. To investigate the presence and distribution of aggregates, we used two photomultiplier tube (PMT) detectors with 405 nm laser excitation for DAPI (Thermo Fisher Scientific, catalog no. P36962) and 488 nm for GFP fluorescence scanned sequentially. Yeast cells were visualized using a 1AU confocal pinhole, and typically 20–25 z-stacks were acquired, each with an optical thickness of 0.28 µm. Cytosolic aggregates were counted manually from the z-stack Maximum Intensity Projection merge. The metastable co-localization experiments in the *hsp42-GFP* strain were performed as above, with the additional detection of metastable proteins by immunofluorescence. Metastable proteins were labeled using a standard immunofluorescence protocol, with a 1:1000 dilution of the FLAG antibody (Sigma, catalog no. F1804) and 1:500 dilution of the Alexa Fluor 568 secondary antibody (Invitrogen, catalog no. A11031), and mounted with DAPI (Thermo Fisher Scientific, catalog no. P36962; *Figure 4—source data 1*). To analyze the number of RFP and GFP aggregates in *C. elegans*, we recorded fluorescence images of the head region of worms that expressed wrmScarlet (RFP) and GFP in body wall muscles (ACH87 strain). Photographs were captured with a Zeiss 700 laser-scanning confocal microscope using a 40× oil objective (NA 1.3). To investigate the presence and distribution of aggregates, we used two PMT detectors with 488 nm laser excitation for GFP fluorescence and 555 nm excitation for RFP fluorescence, scanned sequentially. Worm head regions were visualized using a 1AU confocal pinhole, and typically 35–45 z-stacks were acquired, each with an optical thickness of 1 µm. 8–15 animals were taken per condition per each day of worms' adulthood, starting from L4 larvae as day 0 until day 5 of adulthood. Next, z-stacks were merged using Maximum Intensity Projection. Finally, RFP and GFP aggregates were extracted from the photographs and counted automatically using ImageJ software (*Figure 5—source data 1*). For all of the microscopy experiments, the exposure parameters were chosen such that no saturation effect of the fluorescent signal was present and kept at the same level during the entire experiment.

## Co-localization analysis

Fluorescence co-localization between the Alexa568 FLAG (red channel) and Hsp42-GFP (green channel) was calculated for individual yeast cells, with three cells per condition (marked as individual regions of interest). Pearson's correlation coefficients above the Costes threshold were calculated using the Co-localization Threshold plugin in ImageJ. This was followed by running the Co-localization Test plugin in ImageJ to test statistical significance of the calculations. A resulting p value of 1.00 indicated statistical significance.

## Aggregate size analysis

Because of the small sizes of the aggregates that were at the resolution limit of the confocal microscope and the large diversity of their intensity in each cell, an automatic particle analysis based on image thresholding was not possible. Therefore, we analyzed the average size of the aggregates using a manual approach of defining aggregate boundaries and measuring the aggregate size with ImageJ software, with n = 10 analyzed for each condition (*Figure 4—source data 1*).

## Worm maintenance and strains

Standard conditions were used for the propagation of *C. elegans* (*Brenner, 1974*). Briefly, the animals were synchronized by hypochlorite bleaching, hatched overnight in M9 buffer (3 g/l $KH_2PO_4$, 6 g/l $Na_2HPO_4$, 5 g/l NaCl, and 1 M $MgSO_4$), and subsequently cultured at 20 °C on nematode growth medium (NGM) plates (1 mM $CaCl_2$, 1 mM $MgSO_4$, 5 µg/ml cholesterol, 25 mM $KPO_4$ buffer pH 6.0, 17 g/l agar, 3 g/l NaCl, and 2.5 g/l peptone) seeded with the *Escherichia coli* HB101 or OP50 strain as a food source. The following *C. elegans* strains were used:

> CL2122: dvIs15 [(pPD30.38) unc-54(vector) + (pCL26) mtl-2::GFP]. Control strain for GMC101.
> GMC101: dvIs100 [punc-54::A-beta-1–42::unc-54-3'-UTR+ mtl-2p::GFP]
> ACH87: wacIs11 [pmyo-3::mGFP::SL2 gdp-2-wrmScarlet::unc-54-3'UTR, unc-119(+)]
> OW40: zgIs15 [punc-54::αsyn::YFP]IV (*van Ham et al., 2008*)
> OW450: rmIs126 [punc-54::YFP]V. Control strain for OW40 (*van Ham et al., 2008*)

## Molecular biology for worm studies

An RNAi construct that targeted the *dnj-21* gene was created by PCR amplification of the gene from cDNA pools that were generated from RNA. Primers that were used to generate PCR products were designed to amplify the full cDNA sequence of *dnj-21*. The PCR product was digested with XhoI and KpnI restriction enzymes and cloned into the XhoI and KpnI-digested L4440 vector. Constructs for the expression of wrmScarlet (RFP; mCherry derivative) (*El Mouridi et al., 2017*) and mGFP in body wall muscles of worms were created using the SLiCE method (*Zhang et al., 2012*) and the Three Fragment Gateway System (Invitrogen). Briefly, wrmScarlet from the pSEM87 plasmid was PCR amplified using the following primers: ggaaactgcttcaacgcatcatggtcagcaagggagag and gaagagtaattggacttacttgtagagctcgtccatt. The PCR product was used to replace the mKate2 sequence in the pCG150 vector with the pmyo-3::mGFP::SL2 gdp-2-mKate2::unc-54-3'UTR insert. Cloned constructs were sequenced for insert verification.

## Worm transformation

The *C. elegans* ACH87 transgenic strain was created using biolistic bombardment as described previously (*Praitis et al., 2001*). *unc-119* rescue was used as a selection marker.

## Worm RNAi

RNAi was achieved by feeding worms *E. coli* HT115(DE3) bacteria that was transformed with a construct that targeted the *dnj-21* gene. *E. coli* HT115(DE3) that was transformed with the L4440 empty vector was used as a control. LB medium (10 g/l tryptone, 10 g/l NaCl, and 5 g/l yeast extract) was inoculated with transformed bacteria and cultured at 37 °C at 180 rotations per minute. Bacterial culture was induced with 1 mM isopropyl β-D-1-thiogalactopyranoside (IPTG) once the culture $OD_{600}$ reached 0.4–0.6. After 2 hr, the bacteria were pelleted. For liquid culture, the bacterial pellet was added to S-medium and used for further worm culture. For cultures on solid medium, the plates were seeded with bacteria and used after the bacteria dried.

### Worm total protein isolation and western blot

GMC101 L1 larvae were cultured in liquid S-medium for 3 days, first at 20 °C for 24 hr and then at 20 °C for 48 hr or at 22 or 25 °C for 48 hr to induce Aβ aggregation. Total proteins were isolated from frozen worm pellets. Samples were thawed on ice in lysis buffer (20 mM Tris pH 7.4, 200 mM NaCl, and 2 mM PMSF) and sonicated three times for 10 s. The lysate was centrifuged at 2800× *g* for 5 min at 4 °C. The pellet that contained debris was discarded. Protein concentrations were measured by DirectDetect. Proteins were separated by SDS-PAGE (using 15 % gels) or native PAGE (12 % gels) and then transferred to PVDF membranes. Immunodetection was performed according to standard techniques using chemiluminescence. The following antibodies were used: custom-made rabbit antibody against DNJ-21, purified anti-beta amyloid 1–16 antibody (BioLegend, catalog no. 803001), tubulin (Sigma, catalog no. T9026), mCherry (Abcam, catalog no. ab167453), and GFP (Sigma, catalog no. 11814460001).

### Motility assay

The worms were placed in a drop of M9 buffer and allowed to recover for 30 s to avoid the observation of behavior that is associated with stress, after which the number of body bends was counted for 30 s. The experiments were conducted in triplicate. For each experiment, at least 35 worms were analyzed (*Figure 5—source data 2*).

### Automated body bend assay

Automated motility assessment was performed with a tracking device that was developed and described previously (*Perni et al., 2018*). Briefly, worms were washed off the plates with M9 buffer and spread over a 9 cm NGM plate in a final volume of 5 ml, after which their movements were recorded for 120 s. Videos were analyzed in a consistent manner to track worm motility (bends per minute) and swimming speed. The results show the mean ± SEM from two independent experiments (*Figure 5—source data 2*).

### Aβ quantification

Aβ aggregates were calculated by dividing the signal that was detected with anti-Aβ antibody by the protein signal that was detected with Coomassie staining. For each temperature condition, aggregate levels were normalized to the control. The data are expressed as the mean ± SD (n = 3). Overall differences between conditions were assessed using unpaired *t*-tests by assuming unequal variance (*Figure 5—source data 2*).

### Statistical analysis

For the statistical analysis, two-tailed, unpaired *t*-tests were used by assuming equal variance unless otherwise stated. Values of p≤0.05 were considered statistically significant. For the statistical analysis of the confocal microscopy co-localization data, see the co-localization analysis section in the Materials and methods.

## Acknowledgements

We thank Tiago Outeiro, Bernd Bukau, Axel Mogk, Magdalena Boguta, Henrik Bringmann, Thomas Boulin, Ellen Nollen, and Agnieszka Sztyler for materials and experimental assistance and CGC (funded by the National Institutes of Health Office of Research Infrastructure Programs, P40 OD010440) for providing *C. elegans* strains.

## Additional information

#### Competing interests

Agnieszka Chacinska: Reviewing editor, *eLife*. The other authors declare that no competing interests exist.

## Funding

| Funder | Grant reference number | Author |
| --- | --- | --- |
| Foundation for Polish Science | International Research Agendas programme "Regenerative Mechanisms for Health" MAB/2017/2 (co-financed by the European Union under the European Regional Development Fund) | Agnieszka Chacinska |
| National Science Centre | 2015/18/A/NZ1/00025 | Agnieszka Chacinska |
| Polish Ministerial Funds for Science | Ideas Plus programme 000263 | Agnieszka Chacinska |
| Deutsche Forschungsgemeinschaft | Copernicus Award | Agnieszka Chacinska |
| Foundation for Polish Science | Copernicus Award | Agnieszka Chacinska |
| Foundation for Polish Science | HOMING POIR.04.04.00-00-3FE4/17 (co-financed by the European Union under the European Regional Development Fund) | Urszula Nowicka |
| National Science Centre | POLONEZ 2016/23/P/NZ3/03730 (European Union's Horizon 2020 research and innovation programme under the Marie Sklodowska-Curie grant agreement No 665778) | Barbara Uszczynska-Ratajczak |
| National Science Centre | POLONEZ UMO-2016/21JPJNZ3/03891 (European Union's Horizon 2020 research and innovation programme under the Marie Sklodowska-Curie grant agreement No 665778) | Michal Turek |
| EMBO | 7124 | Maria Sladowska |
| William B. Harrison Foundation | | Michele Vendruscolo |
| University of Cambridge | Centre for Misfolding Diseases | Michele Vendruscolo |

The funders had no role in study design, data collection and interpretation, or the decision to submit the work for publication.

## Author contributions

Urszula Nowicka, Conceptualization, Data curation, Formal analysis, Funding acquisition, Investigation, Project administration, Resources, Validation, Visualization, Writing - original draft, Writing – review and editing; Piotr Chroscicki, Conceptualization, Investigation, Resources, Validation, Visualization, Writing – review and editing; Karen Stroobants, Investigation, Resources, Validation, Writing – review and editing; Maria Sladowska, Conceptualization, Formal analysis, Funding acquisition, Resources, Validation, Visualization, Writing – review and editing; Michal Turek, Formal analysis, Funding acquisition, Resources, Visualization, Writing – review and editing; Barbara Uszczynska-Ratajczak, Data curation, Formal analysis, Funding acquisition, Investigation, Software, Visualization, Writing – review and editing; Rishika Kundra, Investigation, Writing – review and editing; Tomasz Goral, Michele Perni, Formal analysis, Investigation, Writing – review and editing; Christopher M Dobson, Conceptualization; Michele Vendruscolo, Conceptualization, Funding acquisition, Supervision, Writing – review and

editing; Agnieszka Chacinska, Conceptualization, Funding acquisition, Project administration, Supervision, Writing – review and editing

### Author ORCIDs
Urszula Nowicka  http://orcid.org/0000-0001-7446-3786
Piotr Chroscicki  http://orcid.org/0000-0002-4291-3284
Michele Perni  http://orcid.org/0000-0001-7593-8376
Michele Vendruscolo  http://orcid.org/0000-0002-3616-1610
Agnieszka Chacinska  http://orcid.org/0000-0002-2832-2568

### Decision letter and Author response
Decision letter https://doi.org/10.7554/eLife.65484.sa1
Author response https://doi.org/10.7554/eLife.65484.sa2

## Additional files

### Supplementary files
• Supplementary file 1. Side-by-side comparison of gene expression levels that were attributable to heat shock and the *pam16-3* mutant.
• Supplementary file 2. Nucleotides used to clone protein coding sequences and CUP1 promoter into pESC-URA.
• Supplementary file 3. List of plasmids used in this study.
• Transparent reporting form

### Data availability
Sequencing data have been deposited in GEO under accession codes GSE147284.

The following dataset was generated:

| Author(s) | Year | Dataset title | Dataset URL | Database and Identifier |
|---|---|---|---|---|
| Stroobants K, Uszczynska-Ratajczak B, Kundra R, Nowicka U, Chroscicki P, Chacinska A, Vendruscolo M | 2020 | Cytosolic aggregation of mitochondrial proteins enhances degeneration of cellular homeostasis | https://www.ncbi.nlm.nih.gov/geo/query/acc.cgi?acc=GSE147284 | NCBI Gene Expression Omnibus, GSE147284 |

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
