## [Decision Letter]

**Acceptance summary:**

**Decision letter after peer review:**

Thank you for submitting your article "Cytosolic aggregation of mitochondrial proteins disrupts cellular homeostasis by stimulating other proteins aggregation" for consideration by *eLife*. Your article has been reviewed by 3 peer reviewers, one of whom is a member of our Board of Reviewing Editors, and the evaluation has been overseen by David Ron as the Senior Editor. The following individual involved in review of your submission has agreed to reveal their identity: Chris Meisinger (Reviewer #2).

Essential Revisions:

– Section II results – The authors use a temperature sensitive allele so the results must FIRST be normalized to temperature shift alone and no results should be discussed in this section without this normalization.

– In the aggregation assays the authors also find mature, i.e. processed mitochondrial proteins in the pellet fraction (e.g. Figure 1A). These proteins seem to be derived from mitochondria (because this processing occurs only within the organelle). Is there an explanation for their presence in the pellet? Why should mature proteins aggregate? How efficient were mitochondria lysed before the spin in the aggregation assay? Could there still be intact mitochondria present, which would also pellet? A further important control here would be to use isolated mitochondria from the overexpressing strains and validate that the precursors do indeed not reside within the organelle. A separation of mitochondrial and cytosolic fractions would also be helpful to clarify this.

Did the authors analyze also cells without overexpression of the precursor proteins to investigate the behavior of the endogenous proteins?

– Figure 1B: can overexpression of Hsp104 and/or Hsp42 rescue the phenotype here? Figure 3J. pRip aggregates increase in the absence of Hsp104. Then what about the effect of the absence of Hsp42? Overexpression of Hsp104 and Hsp42, in turn, suppresses pRip1 aggregate?

– A loading control is required in Figure 5E and in the suppl. of Figure 5 D-E. Also please explain how was Abeta quantified here.

Textual Changes

– Results section I – Define precursor as (p) once and then use consistently.

– Results section II – Pot1 is not a mitochondrial but rather a peroxisomal protein.

– Define MIA in line 210.

– Hsp42 and Hsp104, the only chaperones upregulated at the protein level, are specific for inclusion bodies – this warrants some mention/discussion. Also they have clearly been implicated in inclusion body and not in aggregate physiology and the correct terminology should be used (for example at later parts Line 327).

– Results section III – "showed the most severe drop in lethality as presented" I am assuming the authors meant drop in viability.

– The results in *C. elegans* should be moved into their own section. In this section it is not clear what GFP and RFP are fused to or whether they are simply soluble cytosolic molecules (Line 303).

– All Figures: Molecular Weights should be shown for all gels and in the figure legends please state clearly how many independent times each gel was repeated.

– Figure 1 – "We selected these mitochondrial genes from KEGG 97 analysis" - The reason for choosing these mitochondrial genes should be more clearly explained in the main text.

– Fig, 1B – " The tendency to aggregate and its harmful consequences correlated well with growth defects of yeast strains under oxidative conditions when galactose was used as a carbon source." -- Growth conditions for each multi-step centrifugation step assay should be shown. For example, do cells grown in glycerol and in galactose show similar tendencies of aggregate formation?

– Figure 1B – "When glucose was used as a carbon source, a gain of stress resistance was observed upon the overproduction of Atp20, suggesting protective mechanism stimulation" – Can the authors state this only based on Figure 1B?

– Figure 1F – Why did the amount of insoluble pRip1 (P-125K) decrease to half at 5 h as compared with 3h while the total amount of pRip1 even increased?

– Figure 1A, E, F – Why does the amount of mRip1 in S^-1^25K look so different between A and E/F?

– Figure 2: gene names in yeast should appear in Italics.

– 2A – The authors should clearly state if the control cells were also grown in copper.

– 2B – Data on pam16-3 should be shown relative to control cells at the same "restrictive" temp.

– 2C – YLR413W is now called INA1.

– Figure 2C – Explanation for how POT1, DLD3, AGP3, and PDC6 are associated with mitochondrial functions is necessary.

– Figure 3A – Hsp104 protein levels were increased at permissive temperature in pam16-1 and pam18-1 mutant cells in the presence of CCCP. Then what about their transcripts at permissive temperature in pam16-3 mutant cells?

– Figure 3G: what is the second band in the immunoblot of Hsp42 in the second lane?

– The changes in transcript levels are sometimes not so large, although statistically being significant. Hence, please clearly describe how you decided on "the affected genes" and clearly state the reasons for choosing specific proteins for follow up.

– Since it is not clear if the observed changes in transcriptome and protein levels are direct consequences of accumulated precursor protein aggregates or indirect effects (For example, the effects on e.g. ER chaperones like Jem1 should be indirect) please word this more clearly.

– Figure 3K-"Hsp104 protein came in the pellet fraction in higher amount than when compared to the soluble fraction in our aggregation assay, suggesting that Hsp104 bound present their aggregates" -- However, Hsp104 can be found in pellet not in sup fractions even in the absence of CCCP. This means that Hsp104 is always mainly in the pellet, but was just induced by CCCP without any increased tendency to go to the pellet. Please discuss

– Figure 4A-C – Why do the effects of overexpression of different proteins differ between pRip1 and pSod2? Please discuss.

– Figure 4A-C – The protein levels of Sod2 look similar between Atp2 and Cox8 overexpression in A (on the gel), but significantly different in C (by quantification). The amounts of pRip1 look similar between vector and Atp20 in A (on the gel), but significantly different in B (by quantification). Please ensure that your quantification is correct.

- Figure 4D – "The accumulated precursor proteins also co-aggregated with metastable proteins in the insoluble fraction based on the aggregation assay analysis" -- This may be misleading. mRip1 was found in insoluble fractions in the absence or presence of Atp2 overexpression and therefore induced pRip1 may just behave like mRip1 in the presence of Atp2 overexpression. A straightforward interpretation would be "Rip1 co-aggregated with metastable proteins in the insoluble fraction based on the aggregation assay analysis".

– Figure 4F, G – It may not be appropriate to discuss the tendency of aggregate formation solely on the basis of just one or two differences in the number of a-Syn aggregates, ignoring the sizes of the aggregates. Besides, the numbers of aSyn-WT and sSyn-A53T aggregates increased in pam16-3 mutant cells at 19{degree sign}C by 30% or so, but not at 37{degree sign}C. The interpretation of this observation that the aggregates sizes increase at 37{degree sign}C is not experimentally supported. The explanation of the use of A53T mutant is also necessary.

– Figure 4H, I – "A higher accumulation of pRip at combined conditions of the pam16-3 stimulated mitochondrial import defect and α-Syn aggregation at 37{degree sign}C"-This is not evident.

– In the Discussion – the story here complements very nicely recent findings that non-processed (but imported!) precursor proteins aggregate in the mitochondrial matrix and initiate an mtUPR like response (actually with transcriptional upregulation of very similar cytosolic chaperones as found here (see Poveda-Huertes, Mol Cell 2020)). Could there be a link (e.g. complementary manner) of the two pathways? The paper should be included in the references, in particular because it demonstrated for the first time that non-processed, immature mitochondrial precursor proteins are prone to aggregation.

Text editing

The writing is still quite raw and requires more polishing to fit the journal. Specifically, the writing is often not streamlined and convoluted. Each figure should correspond to a Results section and should have its own section header. There are many typos and unnecessary use of the word "the" (few examples below).

Typos:

Line 137: similarly as at the condition when Rip1 was overproduced.

Line 150: at this conditions.

Line 166: pRpi1 production for.

Line 194: that with extended of time the.

Line 195: 'ABS-transporter' should read 'ABC transporters'.

Line 197: which gene levels also increased.

Line 205: for most of the them.

Line 230: upregulation of a specific molecular chaperones.

Line 249: bound present there aggregates.

Line 297: proteins aggregation.

Line 301: which stimulated mitochondrial import defect.

Less use of "the":

Line 274: tagged with the GFP.

Line 274/5: By the confocal microscopy experiments.

Line 276: in response to the mitochondrial.

Line 303: aggregation of the RFP and GFP.

*Reviewer #1:*

The article by Nowicka et al. aims to explore the potential role of mitochondrial dysfunctions, including mitochondrial import defects, in contributing to the progression of neurodegenerative diseases. Building on previous publications that in Alzheimer's disease patients there is a transcriptional down regulation of proteins involved in oxidative phosphorylation, they use the yeast homologues of these proteins to show the consequences of their cytosolic accumulation.

Specifically, they find at upon mitochondrial import defects (genetic or chemical) or upon overexpression, a subset of mitochondrial precursors, accumulate in the cytosol where they form insoluble aggregates. These, in turn, both trigger a cytosolic chaperone response as well as stimulate the cytosolic aggregation of other mitochondrial proteins. This also then causes the downstream aggregation of non-mitochondrial proteins including model substrates known to play a role in neurodegeneration. This had a drastic impact on cytosolic proteostasis. Nicely, they show that this is conserved to worms and employ two disease relevant models of aggregation – α synuclein and Abeta aggregation.

The topic of this manuscript is extremely important and the findings are interesting. This is the right time to be exploring these questions as the impact of mistargeting of mitochondrial precursors is becoming better and better understood.

The writing is still quite raw and requires more polishing to fit the journal. Specifically, the writing is often not streamlined and convoluted. Each figure should correspond to a Results section and should have its own section header. There are many typos and unnecessary use of the word "the" (few examples below)

Typos:

Line 137: similarly as at the condition when Rip1 was overproduced

Line 150: at this conditions

Line 166: pRpi1 production for

Line 194: that with extended of time the

Line 197: which gene levels also increased

Line 205: for most of the them

Line 230: upregulation of a specific molecular chaperones

Line 249: bound present there aggregates

Line 297: proteins aggregation

Line 301: which stimulated mitochondrial import defect

Less use of "the":

Line 274: tagged with the GFP

Line 274/5: By the confocal microscopy experiments

Line 276: in response to the mitochondrial

Line 303: aggregation of the RFP and GFP

More specific comments on the data/writing itself:

Results

Section I – Define precursor as (p) once and then use consistently

Section II – Pot1 is not a mitochondrial but rather a peroxisomal protein

– The authors use a temperature sensitive allele so the results must FIRST be normalized to temperature shift alone and no results should be discussed in this section without this normalization.

– Define MIA in line 210

– Hsp42 and Hsp104, the only chaperones upregulated at the protein level, are specific for inclusion bodies – this warrants some mention/discussion. Also they have clearly been implicated in inclusion body and not in aggregate physiology and the correct terminology should be used (for example at later parts Line 327)

Section III – "showed the most severe drop in lethality as presented" I am assuming the authors meant drop in viability

– The results in *C. elegans* should be moved into their own section. In this section it is not clear what GFP and RFP are fused to or wether they are simply soluble cytosolic molecules (Line 303)

All Figures: Molecular Weights should be shown for all gels and in the figure legends please state clearly how many independent times each gel was repeated.

Figure 2: gene names in yeast should appear in Italics

2A – The authors should clearly state if the control cells were also grown in copper

2B – Data on pam16-3 should be shown relative to control cells at the same "restrictive" temp.

2C – YLR413W is now called INA1

*Reviewer #2:*

Impairment of the mitochondrial protein import machinery leads to accumulation of precursor proteins destined to mitochondria in the cytosol. To cope with this burden a variety of rescue/stress responses were recently discovered to ensure cell viability.

In this study the authors propose a novel proteostasis mechanism in which mitochondrial precursor proteins (when overexpressed) accumulate in the cytosol and are prone to aggregation cause a co-aggregation of other proteins. This is accompanied with an increased expression of cytosolic chaperones, which assist the cell to cope with this overload of protein aggregates. This is a highly interesting topic relevant for a broad audience. Most of the experiments were performed in the model yeast and the quality of the experiments is very good. In principle the mechanistical concept that the authors propose is a very exciting one and several of the findings here point into this direction. However, I have some important points, which the authors should address before recommending publication of this paper in *eLife*.

These are related to the clarification why mature (processed) mitochondrial precursor proteins accumulate in the pellet fraction in the aggregation assay employed here and how efficient mitochondria were lysed here.

Another issue was to clarify if this mechanism is specific for mitochondrial precursor proteins or if it also relates e.g. to ER precursors.

A highly exciting add would be to rebuild this mechanisms in vitro by combining cytosol from precursor overexpressing strains to control lysates and test for snowball co-aggregation.

Impairment of the mitochondrial protein import machinery leads to accumulation of precursor proteins destined to mitochondria in the cytosol. To cope with this burden a variety of rescue/stress responses were recently discovered to ensure cell viability.

In this study the authors propose a novel proteostasis mechanism in which mitochondrial precursor proteins (when overexpressed) accumulate in the cytosol and are prone to aggregation cause a co-aggregation of other proteins. This is accompanied with an increased expression of cytosolic chaperones, which assist the cell to cope with this overload of protein aggregates. This is a highly interesting topic relevant for a broad audience. Most of the experiments were performed in the model yeast and the quality of the experiments is very good. In principle the mechanistical concept that the authors propose is a very exciting one and several of the findings here point into this direction. However, I have some important points, which the authors should address before recommending publication of this paper in *eLife*.

– In the aggregation assays the authors also find mature, i.e. processed mitochondrial proteins in the pellet fraction (e.g. Figure 1A). These proteins seem to be derived from mitochondria (because this processing occurs only within the organelle). Is there an explanation for their presence in the pellet? Why should mature proteins aggregate? How efficient were mitochondria lysed before the spin in the aggregation assay? Could there still be intact mitochondria present, which would also pellet?

A further important control here would be to use isolated mitochondria from the overexpressing strains and validate that the precursors do indeed not reside within the organelle. A separation of mitochondrial and cytosolic fractions would also be helpful to clarify this.

Did the authors analyze also cells without overexpression of the precursor proteins to investigate the behavior of the endogenous proteins?

– Would also overexpressed ER (or from other cell compartments) proteins aggregate (maybe also dependent on the signal sequence?)? If not, what is the idea/hypothesis that this specific for a mitochondrial precursor?

– Figure 1B: can overexpression of Hsp104 and/or Hsp42 rescue the phenotype here?

– the story here complements very nicely recent findings that non-processed (but imported!) precursor proteins aggregate in the mitochondrial matrix and initiate an mtUPR like response (actually with transcriptional upregulation of very similar cytosolic chaperones as found here (see Poveda-Huertes, Mol Cell 2020)). Could there be a link (e.g. complementary manner) of the two pathways? The paper should be included in the references, in particular because it demonstrated for the first time that non-processed, immature mitochondrial precursor proteins are prone to aggregation.

– *C. elegans* experiments (Figure 5): In my opinion these experiments do not contribute much here. I am also missing experiments, which show that the dnj21 worms have an impaired mitochondrial protein import. I would leave this part rather out. If it stays in the manuscript it would be good to have a loading control in Figure 5E and to clarify what happens with mitochondrial precursor proteins in the cytosol (do they accumulate in this system as well? And if yes, do they also aggregate like in yeast or is the concentration just too low?; what happen with C. elegans 'high risk' precursor here?). Also in the suppl. of Figure 5 D-E: how was Abeta quantified here? Showing a loading control would be good.

– An important add to this paper would be an in vitro assay in the yeast model employed here, showing the proposed 'snowball effect' of protein aggregation: Mixing cytosol from overexpressing strains with cytosol of a non-overexpressing strain: would there also be a co-aggregation? This would mark the paper as a very important milestone paper in the field (if this is technically doable).

– Figure 3G: what is the second band in the immunoblot of Hsp42 in the second lane?

*Reviewer #3:*

Mitochondrial protein import is essential for eukaryotic cell viability. When it is compromised, precursor forms of mitochondrial proteins will accumulate in the cytosol as well as at the level of mitochondrial import channels. Previous studies including those from the authors' laboratory showed that, since accumulated mitochondrial protein precursors may form toxic aggregates, cells have a mechanism to respond to and cope with them properly. In this manuscript, Nowicka et al. analyzed the effects of compromised mitochondrial import in yeast and *C. elegans* by using mutations in import machineries, overexpression of mitochondrial proteins, and CCCP dissipating the mitochondrial membrane potential. They found that compromised mitochondrial protein import caused global changes in transcriptome and protein levels of especially, chaperones including Hsp104 and Hsp42, ABC transporters, and mitochondrial proteins, which may lead to growth defects (yeast) and decreased motility (*C. elegans*). The obtained results could offer a basis for future investigation and idea on the entire picture of the mitochondrial import defect response.

My impression is that the work, as it stands, is rich in interesting data and hints, yet it remains descriptive and therefore premature.

1) The authors used three different ways to impair mitochondrial protein import, which led to precursor accumulation. However, of course, the compromised level of protein import differs for the different methods, and therefore it is not easy to compare the effects among these three methods properly and to delineate the essence of the effects common to these three methods. Indeed the observed changes caused by different methods to impair mitochondrial protein import are not the same, which is at the moment, difficult to reconcile. Besides, the import defects caused by the temperature-sensitive (ts) mutations could pause the effects of temperature changes, as well, and there are some import defects even at permissive temperature, so that interpretation of the results is complicated, as the authors noted.

2) The authors analyzed the changes in transcriptome and protein levels, but the way to analyze them are not consistent with each other. The former analysis handles the entire transcriptome, but the latter analysis assesses a limited number of selected proteins. The changes in transcript levels are sometimes not so large, although statistically being significant. How to pick up "the affected genes" is not clearly described. The protein-level changes are not analyzed in a non-biased manner due to lack of available antibodies, which is understandable, but at least the reason for choosing some proteins should be clearly described. Besides, if the authors analyze only a limited number of proteins, I would like to see more detailed analyses of the proteins of interest. For example, 'the post-transcriptional changes' of some genes are presumably ascribed to the changes in degradation, but then this should be directly tested by cycloheximide chase experiments. The conditions to impair protein import are not the same between the transcriptome analyses and protein level analyses, which hampers the direct comparison between the two classes of analyses.

3) The most serious problem in this kind of analysis would be that it is not clear if the observed changes in transcriptome and protein levels are direct consequences of accumulated precursor protein aggregates or indirect effects. For example, the effects on e.g. ER chaperones like Jem1 should be indirect.

4) The apparent involvement of Hsp104 and Hsp42 in the response to accumulated precursor protein aggregates could be interesting, but no clue to the mechanisms of their induction and their specific (?) association with aggregates is obtained.

5) The logic of the text flow is sometimes hard to follow and the presentation of the figure panels is often not reader-friendly.

Specific points.

Figure 1 – "We selected these mitochondrial genes from KEGG 97 analysis" --The reason for choosing these mitochondrial genes should be more clearly explained in the main text.

Fig, 1B – " The tendency to aggregate and its harmful consequences correlated well with growth defects of yeast strains under oxidative conditions when galactose was used as a carbon source." -- Growth conditions for each multi-step centrifugation step assay should be shown. For example, cells grown in glycerol and in galactose show similar tendencies of aggregate formation?

Figure 1B – "When glucose was used as a carbon source, a gain of stress resistance was observed upon the overproduction of Atp20, suggesting protective mechanism stimulation".

– Can the authors state this only based on Figure 1B?

Figure 1C – "We also extended the analysis for the precursor form of the other proteins for which antibodies were available at the laboratory and allowed for precursor forms detection, namely: the mitochondrial matrix superoxide dismutase (Sod2), and mitochondrial malate dehydrogenase 1 (Mdh1)" – Why did the authors choose only Sod2 and Mdh1 for further analyses?

Figure 1F – Why did the amount of insoluble pRip1 (P-125K) decrease to half at 5 h as compared with 3h while the total amount of pRip1 even increased?

Figure 1A, E, F – Why does the amount of mRip1 in S^-1^25K look so different between A and E/F?

Figure 2C – Explanation for how POT1, DLD3, AGP3, and PDC6 are associated with mitochondrial functions is necessary.

Line 195 – 'ABS-transporter' should read 'ABC transporters'.

Figure 3A – Hsp104 protein levels were increased at permissive temperature in pam16-1 and pam18-1 mutant cells in the presence of CCCP. Then what about their transcripts at permissive temperature in pam16-3 mutant cells?

Figure 3I – pRip1-FLAG and Hsp42-Alexa appear to co-localize. Then what about Hsp104? "These deposits co-localised with the GFP signals and indicated that Hsp42 sequestered pRip1 aggregates." -- The possible physical interactions should be experimentally tested.

Figure 3J. pRip aggregates increase in the absence of Hsp104. Then what about the effect of the absence of Hsp42? Overexpression of Hsp104 and Hsp42, in turn, suppresses pRip1 aggregate?

Figure 3K-"Hsp104 protein came in the pellet fraction in higher amount than when compared to the soluble fraction in our aggregation assay, suggesting that Hsp104 bound present their aggregates" -- However, Hsp104 can be found in pellet not in sup fractions even in the absence of CCCP. This means that Hsp104 is always mainly in the pellet, but was just induced by CCCP without any increased tendency to go to the pellet.

Figure 4A-C – Why do the effects of overexpression of different proteins differ between pRip1 and pSod2?

Figure 4A-C – The protein levels of Sod2 look similar between Atp2 and Cox8 overexpression in A (on the gel), but significantly different in C (by quantification). The amounts of pRip1 look similar between vector and Atp20 in A (on the gel), but significantly different in B (by quantification). Does the quantification correct?

Figure 4D – "The accumulated precursor proteins also co-aggregated with metastable proteins in the insoluble fraction based on the aggregation assay analysis" -- This may be misleading. mRip1 was found in insoluble fractions in the absence or presence of Atp2 overexpression and therefore induced pRip1 may just behave like mRip1 in the presence of Atp2 overexpression. A straightforward interpretation would be "Rip1 co-aggregated with metastable proteins in the insoluble fraction based on the aggregation assay analysis"

Figure 4F, G – It may not be appropriate to discuss the tendency of aggregate formation solely on the basis of just one or two differences in the number of a-Syn aggregates, ignoring the sizes of the aggregates. Besides, the numbers of aSyn-WT and sSyn-A53T aggregates increased in pam16-3 mutant cells at 19{degree sign}C by 30% or so, but not at 37{degree sign}C. The interpretation of this observation that the aggregates sizes increase at 37{degree sign}C is not experimentally supported. The explanation of the use of A53T mutant is also necessary.

Figure 4H, I – "A higher accumulation of pRip at combined conditions of the pam16-3 stimulated mitochondrial import defect and α-Syn aggregation at 37{degree sign}C"-This is not evident.

Figure 5E – Loading controls should be shown.

---

## [Author Response]

Essential Revisions:– Section II results – The authors use a temperature sensitive allele so the results must FIRST be normalized to temperature shift alone and no results should be discussed in this section without this normalization.

We fully agree on this point, and indeed all the presented RNA sequencing data for the temperature sensitive allele were normalized to the WT at the corresponding time point of heat shock exposure. Such normalization allowed us to investigate the effect of mutation by minimizing at the same time the effect of stress response triggered by high temperature.

The Reviewers’ comment brought to our attention the fact that we were not clear enough in the way how the data in Section II were presented. Therefore, to better explain the data, we have included a new paragraph describing the temperature effect for the WT and *pam16-3* samples (new Figure2—figure supplement 2). Please note, that for *pam16-3* samples the measured changes in such analysis reflect the mutation and temperature effects. We thus modified the labels in Figure 2 and Figure2—figure supplement 1. We have also added a summary table – Supplementary file 1, showing the impact of temperature and the mutation on the expression of each gene.

– In the aggregation assays the authors also find mature, i.e. processed mitochondrial proteins in the pellet fraction (e.g. Figure 1A). These proteins seem to be derived from mitochondria (because this processing occurs only within the organelle). Is there an explanation for their presence in the pellet? Why should mature proteins aggregate? How efficient were mitochondria lysed before the spin in the aggregation assay? Could there still be intact mitochondria present, which would also pellet? A further important control here would be to use isolated mitochondria from the overexpressing strains and validate that the precursors do indeed not reside within the organelle. A separation of mitochondrial and cytosolic fractions would also be helpful to clarify this.Did the authors analyze also cells without overexpression of the precursor proteins to investigate the behavior of the endogenous proteins?

As suggested by the Reviewers, the mature proteins are indeed present in the pellet fraction. As we indicated in the Figure 1 —figure supplement 2A, this fraction also includes the unbroken cells. We supported this conclusion by the observation that when higher amounts of detergents are used for lysis buffer, a significant drop in the amount of the mature forms of proteins is observed, as shown in Author response image 1.

**Author response image 1. sa2fig1:** Addition of SDS in the lysis buffers results in better cell lysis and smaller amounts of mature forms of proteins in the P4k fraction based on the aggregation assay.

In addition, when we performed the fractionation experiment, we observed that the precursor form of Rip1 (pRip1) is present only in the cytosol (supernatant – S), and the mature form of Rip1 is present in the mitochondrial fraction (M). Still some small amounts of pRip1 are present in the M fraction, likely due to contamination of the M fraction with the S fraction, since small amounts of Hsp104 are also present in the M fraction (see Author response image 2).

**Author response image 2. sa2fig2:** Fractionation of cells expressing pRip1. T- total cells, S – supernatant, cytosolic fraction, M – mitochondrial fraction.

Furthermore, we did analyse the behaviour of the endogenous proteins without overexpression of the precursor forms. In these experiments we used CCCP or dysfunctional mutants in order to observe the precursor forms on the endogenous level. These data are shown in Figure 1E-F in the manuscript.

– Figure 1B: can overexpression of Hsp104 and/or Hsp42 rescue the phenotype here? Figure 3J. pRip aggregates increase in the absence of Hsp104. Then what about the effect of the absence of Hsp42? Overexpression of Hsp104 and Hsp42, in turn, suppresses pRip1 aggregate?

To address this point, we have now included in the revised manuscript a new Results section that discuss the effects of Hsp42 absence and overexpression of Hsp42 and Hsp104. The new results are presented in Figure 3 L-N and Figure 3 —figure supplement 2.

The new data show that Hsp42 deletion does not enhance the accumulation of precursor proteins (new Figure 3 —figure supplement 2A-B). This effect indicates that a lack of Hsp42 is compensated by an upregulation of Hsp104 (new data presented in Figure 3L-M). Moreover, overexpression of Hsp42 did not result in the rescue of the mitochondrial precursor proteins accumulation (new Figure 3 —figure supplement 2E-F). Only Hsp104 overexpression resulted in a decrease in the levels of mitochondrial precursor proteins (new Figure 3N and Figure 3 —figure supplement 2C-D). We have also included the data presented in Figure 3 —figure supplement 2G, showing that overexpression of neither Hsp42 nor Hsp104 did rescue the phenotype presented in Figure 1B.

– A loading control is required in Figure 5E and in the suppl. of Figure 5 D-E. Also please explain how was Abeta quantified here.

A Coomassie staining of proteins served as a loading control for data presented both in Figure 5E and the Figure 5 —figure supplement 1D and 1E (currently Figure 5 —figure supplement 1E and 1G). We have included loading controls in Figure 5 —figure supplement 2D and 2F for both sets of data. We have also included an explanation on how the Aβ quantification was performed in Materials and methods sections: “Aβ aggregates were calculated by dividing the signal that was detected with anti-Aβ antibody by the protein signal that was detected with Coomassie staining. For each temperature condition, aggregate levels were normalized to the control. The data are expressed as the mean ± SD (n = 3). Overall differences between conditions were assessed using unpaired t-tests by assuming unequal variance.”

Textual Changes– Results section I – Define precursor as (p) once and then use consistently.

This change has been made and the short version was used in the text whenever possible.

– Results section II – Pot1 is not a mitochondrial but rather a peroxisomal protein.

We have added an appropriate description in the text.

– Define MIA in line 210.

This change was introduced: “We then tested whether Hsp104 was upregulated in response to mitochondrial import defects that were caused by the pam16-3 mutation, along with the established pam16-1 and pam18-1 mutants and mitochondrial intermembrane space import and assembly (MIA) import pathway mutations mia40-4int and mia40-3 (Wrobel et al., 2015).”

– Hsp42 and Hsp104, the only chaperones upregulated at the protein level, are specific for inclusion bodies – this warrants some mention/discussion. Also they have clearly been implicated in inclusion body and not in aggregate physiology and the correct terminology should be used (for example at later parts Line 327).

We have clarified this point as indicated, and we also adopted this terminology within the paper.

– Results section III – "showed the most severe drop in lethality as presented" I am assuming the authors meant drop in viability.

Indeed, a correction has been introduced to the text.

– The results in *C. elegans* should be moved into their own section. In this section it is not clear what GFP and RFP are fused to or whether they are simply soluble cytosolic molecules (Line 303).

We have moved all *C. elegans* results to a new section. We also added a clear explanation that both GFP and RFP have been used as model cytosolic proteins.

– All Figures: Molecular Weights should be shown for all gels and in the figure legends please state clearly how many independent times each gel was repeated.

We made appropriate changes in the figures and figure legends.

– Figure 1 – "We selected these mitochondrial genes from KEGG 97 analysis" -The reason for choosing these mitochondrial genes should be more clearly explained in the main text.

We have added a new explanation of the gene selection process at the beginning of section: “Metastable mitochondrial precursor proteins can aggregate in the cytosol”.

– Fig, 1B – " The tendency to aggregate and its harmful consequences correlated well with growth defects of yeast strains under oxidative conditions when galactose was used as a carbon source." - Growth conditions for each multi-step centrifugation step assay should be shown. For example, do cells grown in glycerol and in galactose show similar tendencies of aggregate formation?

We have optimized the conditions at which the precursors can be best observed and their fate followed (see Author response image 3). Based on this analysis, we concluded that the most pronounced precursor forms are observed when 2% sucrose is used as a carbon source and therefore for the majority our studies we have used these growth conditions. The only exceptions are experiments with overexpression of metastable proteins, and aggregation assay followed by MG132 treatment, where 2% galactose with 0.1% glucose and 2% galactose was used, respectively. Furthermore, based on these results, whenever the carbon source used allows for precursor forms observation their tendencies to form aggregates is similar, as observed for pRip1 in Figure 1B (2% galactose with 0.1% glucose) vs. Figure 1D (2% sucrose).

**Author response image 3. sa2fig3:** pRip1 levels for the different sources of carbon used for growth conditions. YPG – glycerol based, YPS – sucrose based, YPGal – galactose based, YPGal + 0.1% Glucose – galactose based with addition of glucose.

The growth conditions have been described in the Materials and methods section, and as suggested by the Reviewers, the carbon source type was added in the figure legend for each aggregation assay.

– Figure 1B – "When glucose was used as a carbon source, a gain of stress resistance was observed upon the overproduction of Atp20, suggesting protective mechanism stimulation".– Can the authors state this only based on Figure 1B?

We agree that results of this experiment are not sufficient to make such a statement. Therefore, we have removed it from the manuscript.

– Figure 1F – Why did the amount of insoluble pRip1 (P-125K) decrease to half at 5 h as compared with 3h while the total amount of pRip1 even increased?

Indeed, it seems that there is a decrease of the pRip1 at the 5h time point when compared to the 3h time point. This observation suggests that the longer *pam16-3* was exposed to high temperature, the greater was the effect of the mutation (these cells exhibit slower growth), which might result in more of aggregating proteins present in P4k fraction. Additionally, under such conditions, secondary effects such as autophagy might take place.

– Figure 1A, E, F – Why does the amount of mRip1 in S^-1^25K look so different between A and E/F?

These observations are consistent between the biological replicates. The main difference between experiments presented in Figure 1A vs. Figure 1E/F is that in Figure 1A we are following the overexpression of the precursor form of the Rip1, while in the Figure 1E/F we are following endogenous levels of all Rip1 protein forms (precursor, intermediate, mature). Also, since we produce more protein with overexpression more of it can end up in P4K fraction instead of being fully available for further centrifugation steps.

– Figure 2: gene names in yeast should appear in Italics

We have made the change in the Figure 2 as suggested.

– 2A – The authors should clearly state if the control cells were also grown in copper

We have now clearly stated the information that the control cells were grown when copper was present. This information is now emphasised both in Figure 2A legend and Materials and methods section.

– 2B – Data on pam16-3 should be shown relative to control cells at the same "restrictive" temp.

The data in Figure 2B were shown relative to control cells at the same restrictive temperature. However, as we have mentioned in the response to the Essential Revisions for Section II, we tried to significantly improve the data presentation and description for this section. We apologize that our original data presentation caused a confusion.

– 2C – YLR413W is now called INA1

We have made an appropriate change in Figure 2C.

– Figure 2C – Explanation for how POT1, DLD3, AGP3, and PDC6 are associated with mitochondrial functions is necessary.

We have added an appropriate description for each of the genes, to briefly explain its function.

– Figure 3A – Hsp104 protein levels were increased at permissive temperature in pam16-1 and pam18-1 mutant cells in the presence of CCCP. Then what about their transcripts at permissive temperature in pam16-3 mutant cells?

Based on our RNA-seq data we did not observe any differences in the Hsp104 gene expression between the WT and pam*16-3* at permissive temperature (please see Author response image 4). Hsp104 levels in *pam16-3* and *pam18-1* mutation related experiments were measured without CCCP treatment.

**Author response image 4. sa2fig4:** The change in the transcript per million between the WT and *pam16-3* under permissive temperature.

– Figure 3G: what is the second band in the immunoblot of Hsp42 in the second lane?

This is a non-specific band. We observe it whenever the Atp2_FLAG_ is over-expressed and the detection is made with the GFP antibody (including strains without any GFP). We have marked it with an asterisk to improve the data presentation.

– The changes in transcript levels are sometimes not so large, although statistically being significant. Hence, please clearly describe how you decided on "the affected genes" and clearly state the reasons for choosing specific proteins for follow up.

In the case of pRip1 samples, all detected differentially expressed genes have been reported. According to our analysis only 13 genes in this set show 2-fold expression change (log2FC +/- 1) with statistical significance (FDR < 5%).

For *pam16-3* samples, after performing the global expression profiling followed by the KEGG enrichment analysis, we selected the genes with statistically significant expression changes, FDR<5%. Next, we identified groups of non-mitochondrial genes (>= 10) that share a similar function/role in the cell based on UniProt, SGD and manual literature searches. We identified mitoCPR associated genes and genes coding for chaperones. The mitoCPR pathways were described in a paper Weidberg and Amon, Science 2018, thus we have focused on the genes coding for chaperones. This aspect was also particularly interesting in the context of our finding, showing that mitochondrial precursor proteins can form aggregates. We wanted to study the effects on protein levels of as many molecular chaperones as possible; however due to limited availability of antibodies, we could only focus on selected ones. In this group, the protein level changes were observed for Hsp42 and Hsp104. Therefore, we focused on these two molecular chaperones in the subsequent studies. We improved the description of the protein selection process in the text, including the Materials and methods section.

– Since it is not clear if the observed changes in transcriptome and protein levels are direct consequences of accumulated precursor protein aggregates or indirect effects (For example, the effects on e.g. ER chaperones like Jem1 should be indirect) please word this more clearly.

To be clearer about such a possibility we added the following statement: “The effect of the pam16-3 mutant could be both direct and indirect when considering the various pathways in which these molecular chaperones are involved.”

– Figure 3K-"Hsp104 protein came in the pellet fraction in higher amount than when compared to the soluble fraction in our aggregation assay, suggesting that Hsp104 bound present their aggregates" - However, Hsp104 can be found in pellet not in sup fractions even in the absence of CCCP. This means that Hsp104 is always mainly in the pellet, but was just induced by CCCP without any increased tendency to go to the pellet. Please discuss

We fully agree with this data interpretation, therefore we modified our statement.

– Figure 4A-C – Why do the effects of overexpression of different proteins differ between pRip1 and pSod2? Please discuss

Although we did not address this difference experimentally, we hypothesise they might be justified by differences in: (1) the rate of import into the TOM complex, (2) the protein sequence that consequently might cause differences in biophysical properties affecting their aggregation dynamics, and (3) overall protein abundance. We now included this information in the text.

– Figure 4A-C – The protein levels of Sod2 look similar between Atp2 and Cox8 overexpression in A (on the gel), but significantly different in C (by quantification). The amounts of pRip1 look similar between vector and Atp20 in A (on the gel), but significantly different in B (by quantification). Please ensure that your quantification is correct.

To address this point we have re-examined our data. For the quantification we have used 3 independent biological replicates and the signal was normalized to the loading control (Rpn1 for pSod2 and Pgk1 for pRip1). In Author response image 5, we have provided the amount of pSod2 for Cox8 overexpression for each of the biological replicate. We did not observe any significant change for pRip1 for Atp20. Therefore, it was not marked in original Figure 4B. In Author response image 6, we have provided the amount of pRip1 for Atp2, Cox8, and Atp20 overexpression for each of the biological replicates.

**Author response image 5. sa2fig5:** pSod2 – fold change presented for each biological replicate when Cox8 was overexpressed.

**Author response image 6. sa2fig6:** pRip1 – fold change presented for each biological replicate when Cox8, Atp2, and Atp20 was overexpressed.

- Figure 4D – "The accumulated precursor proteins also co-aggregated with metastable proteins in the insoluble fraction based on the aggregation assay analysis" -- This may be misleading. mRip1 was found in insoluble fractions in the absence or presence of Atp2 overexpression and therefore induced pRip1 may just behave like mRip1 in the presence of Atp2 overexpression. A straightforward interpretation would be "Rip1 co-aggregated with metastable proteins in the insoluble fraction based on the aggregation assay analysis"

We have made a textual change as suggested.

– Figure 4F, G – It may not be appropriate to discuss the tendency of aggregate formation solely on the basis of just one or two differences in the number of a-Syn aggregates, ignoring the sizes of the aggregates. Besides, the numbers of aSyn-WT and sSyn-A53T aggregates increased in pam16-3 mutant cells at 19{degree sign}C by 30% or so, but not at 37{degree sign}C. The interpretation of this observation that the aggregates sizes increase at 37{degree sign}C is not experimentally supported. The explanation of the use of A53T mutant is also necessary.

In our studies, we made an attempt to analyse the changes in aggregate size with the automatic approach. Unfortunately, since the aggregate sizes were at the resolution limit of our confocal microscope and a great diversity of their intensity in each cell, an automatic particle analysis based on image thresholding was not possible. Since indeed we observed the change in the average number of αSyn-WT aggregates per cell for *pam16-3*, at both temperatures, and for αSyn-A53T in *pam16-3* only at 19 °C, but not at 37 °C, we analysed the average size of the aggregates with manual approach of defining aggregates boundaries (Figure4—figure supplement 1F and 1G). Based on our analysis for n=10, there was no difference in the average size of the α-Syn WT aggregates for WT and *pam16-3* cells at both temperatures. However, the average size of the α-Syn A53T aggregates increased at 37 °C in WT and *pam16-3* when compared to 19 °C, with agreement to our hypothesis. As requested, we have included an explanation why A53T mutant was also used.

– Figure 4H, I – "A higher accumulation of pRip at combined conditions of the pam16-3 stimulated mitochondrial import defect and α-Syn aggregation at 37{degree sign}C"-This is not evident.

For the quantification, we have used 3 independent biological replicates and the signal was normalized to the loading control (Pgk1). In Author response image 7, we have provided the amount of pRip1 for each of the biological replicates.

**Author response image 7. sa2fig7:** pRip1 – fold change presented for each biological replicate. Conditions: *pam16-3* at 37°C.

– In the Discussion – the story here complements very nicely recent findings that non-processed (but imported!) precursor proteins aggregate in the mitochondrial matrix and initiate an mtUPR like response (actually with transcriptional upregulation of very similar cytosolic chaperones as found here (see Poveda-Huertes, Mol Cell 2020)). Could there be a link (e.g. complementary manner) of the two pathways? The paper should be included in the references, in particular because it demonstrated for the first time that non-processed, immature mitochondrial precursor proteins are prone to aggregation.

It is an extremely interesting concept of the two responses to complement each other. We have extended the Discussion section by describing such a possibility, and we have included the paper citation in the reference section.

Text editingThe writing is still quite raw and requires more polishing to fit the journal. Specifically, the writing is often not streamlined and convoluted. Each figure should correspond to a Results section and should have its own section header. There are many typos and unnecessary use of the word "the" (few examples below)

We tried to improve the text as much as possible. We also used a native speaker service to improve the language and grammar. As requested, we also introduced headers for each Results sections.

Typos:Line 137: similarly as at the condition when Rip1 was overproduced.

The correction has been made.

Line 150: at this conditions.

The correction has been made.

Line 166: pRpi1 production for.

The correction has been made.

Line 194: that with extended of time the.

The correction has been made.

Line 195: 'ABS-transporter' should read 'ABC transporters'.

The correction has been made.

Line 197: which gene levels also increased.

The correction has been made.

Line 205: for most of the them.

The correction has been made.

Line 230: upregulation of a specific molecular chaperones.

The correction has been made.

Line 249: bound present there aggregates.

The correction has been made.

Line 297: proteins aggregation.

The correction has been made.

Line 301: which stimulated mitochondrial import defect.Less use of "the":Line 274: tagged with the GFP.

The correction has been made.

Line 274/5: By the confocal microscopy experiments.

The correction has been made.

Line 276: in response to the mitochondrial.

The correction has been made.

Line 303: aggregation of the RFP and GFP.

The correction has been made.